# FOURIER TRANSPORTER:
# BI-EQUIVARIANT ROBOTIC MANIPULATION IN 3D

**Haojie Huang, Owen L. Howell[*], Dian Wang[*], Xupeng Zhu[*], Robert Platt[†], Robin Walters[†]**
[*] Equal Contribution, [†] Equally Advising
Northeastern University, Boston, MA 02115, USA
`{huang.haoj; howell.o; wang.dian; zhu.xup; r.walters; r.platt}@northeastern.edu`

## ABSTRACT

Many complex robotic manipulation tasks can be decomposed as a sequence of pick and place actions. Training a robotic agent to learn this sequence over many different starting conditions typically requires many iterations or demonstrations, especially in 3D environments. In this work, we propose Fourier Transporter (FOURTRAN), which leverages the two-fold $SE(d) \times SE(d)$ symmetry in the pick-place problem to achieve much higher sample efficiency. FOURTRAN is an open-loop behavior cloning method trained using expert demonstrations to predict pick-place actions on new configurations. FOURTRAN is constrained by the symmetries of the pick and place actions independently. Our method utilizes a fiber space Fourier transformation that allows for memory-efficient computation. Tests on the RLbench benchmark achieve state-of-the-art results across various tasks.

## 1 INTRODUCTION

Imitation learning for manipulation tasks in $SE(3)$ has emerged as a key topic in robotic learning. Imitation learning is attractive because of its practicality. In contrast to reinforcement learning wherein training happens via a period of autonomous interaction with the environment (which can potentially damage the robot and the environment), imitation learning requires only human demonstrations of the task to be performed which is often safer and easier to provide. Sample efficiency is critical here; the robot needs to learn to perform a task without requiring the human to provide an undue number of demonstrations. Unfortunately, many current state-of-the-art methods are not sample efficient. For example, even after training with one hundred demonstrations, methods like PerAct and RVT still struggle to solve standard RLBench tasks like STACK WINE or STACK CUPS (Shridhar et al., 2023; Goyal et al., 2023).

Why is sample efficiency such a challenge in three dimensions? A big reason is that these tasks are defined over $SE(3)$ action spaces where the policy must output both a 3D position and $SO(3)$ orientation. The orientation component here is a particular challenge for robot learning because $SO(3)$ is not Euclidean and standard convolutional layers, which lack geodesic properties, are not well-adapted for $SO(3)$ convolutions. Discretization of the group $SO(3)$ is also difficult. For example, a grid sampling of 1000 different orientations in $SO(3)$ still only realizes an angular resolution of $36^{\circ}$, which is insufficient for many manipulation tasks. Instead, existing methods, e.g. Shridhar et al. (2023) or Goyal et al. (2023), often fall back on generic self-attention layers that do not take advantage of the geometry of $SO(3)$.

One approach to this problem is to leverage neural-network policy models that are symmetric in $SO(2)$ or $SO(3)$. This has been explored primarily in $SO(2)$ and $SE(2)$ settings, e.g. Wang et al. (2021) and Jia et al. (2023), where significant gains in sample efficiency have been made using policy networks that incorporate steerable convolution kernels (Cohen & Welling, 2017). In $SO(3)$, symmetric models have been mainly applied to pose or descriptor inference, e.g. Klee et al. (2023) and Ryu et al. (2022), but not directly to policy learning. Moreover, most works do not address the dual symmetry present in many pick and place problems, sometimes referred to as *bi-equivariance* (Ryu et al., 2022), where the pick-place action distribution transforms symmetrically (equivariantly)

---

Project website: https://haojhuang.github.io/fourtran_page

when a transformation is applied independently to either the pick or the place pose. This is illustrated in Figure 1. Independent rotations of the gear ($g_1$) and the slot ($g_2$) result in a change ($a' = g_2 a g_1^{-1}$) in the requisite action needed to perform the insertion. While bi-equivariance in policy learning has been studied in $\mathrm{SO}(2)$ (Huang et al., 2023a), models for encoding $\mathrm{SO}(3)$ bi-equivariance in a general policy learning setting have not yet been developed.

This paper proposes *Fourier Transporter* (FOURTRAN), an approach to modeling $\mathrm{SE}(3)$ bi-equivariance using 3D convolutions and a Fourier representation of rotations. Unlike existing methods, e.g. Equivariant Descriptor Fields (Ryu et al., 2022) and TAX-Pose (Pan et al., 2023), our method encodes $\mathrm{SO}(3)$ bi-equivariance inside a general purpose policy learning model rather than relying on point descriptors, which often require sample and optimization during inference. Our key innovation is to parameterize action distributions over $\mathrm{SO}(3)$ in the Fourier domain as coefficients of Wigner $D$-matrix entries. We embed this representation within a 3D translational convolution, thereby enabling us to do convolutions directly $\mathrm{SE}(3)$ without excessive computational cost and with minimal memory requirements. The end result is a policy learning model for imitation learning with high sample efficiency and high angular resolution that can outperform existing $\mathrm{SE}(3)$ methods by significant margins (Table 1). Our contributions are:

- We analyze problems with bi-equivariant symmetry and provide a general theoretical solution to leverage the coupled symmetries.

- We propose Fourier Transporter (FOURTRAN) for leveraging bi-equivariant structure in manipulation pick-place problems in 2D and 3D.

- We achieve state-of-the-art performance on several RLbench tasks (James et al. (2020)). Specifically, FOURTRAN outperforms baselines by a margin of between six percent (STACK-WINE) and *two-hundred* percent (STACK-CUPS).

## 2    RELATED WORK

**Action-centric manipulation.** Traditionally, vision-based manipulation policies (Zhu et al., 2014; Zeng et al., 2017; Deng et al., 2020; Wen et al., 2022) often require pretrained vision models to conduct object detection, segmentation and pose estimation, and may struggle with deformable or granular objects. Action-centric manipulation associates each pixel, voxel, or point with a target position of the end-effector, providing an efficient framework that evaluates a large amount of action with a dense output map. Transporter Networks (Zeng et al., 2021; Seita et al., 2021; Shridhar et al., 2022) combine action-centric representations with end-to-end learning, showing fast convergence speed and strong generalization ability. However, these end-to-end learning methods are often limited to 2D top-down settings and cannot efficiently or accurately solve 3D pick and place tasks.

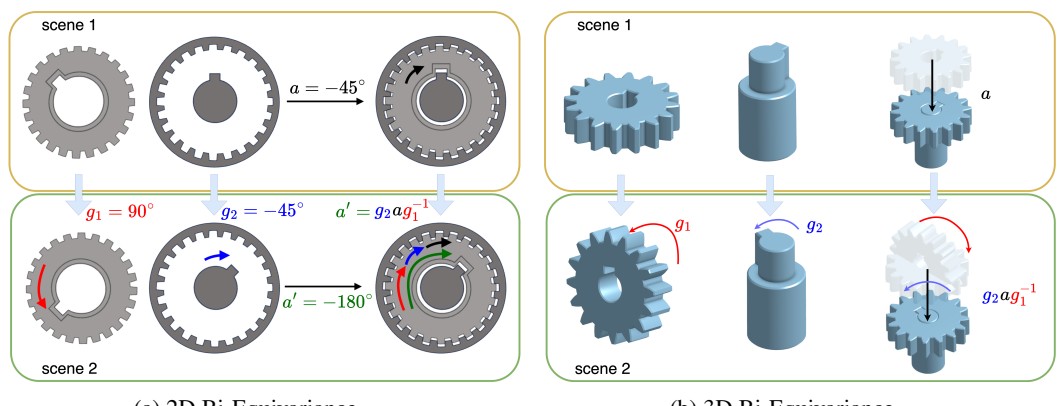

|   |   |
|---|---|
| (a) 2D Bi-Equivariance | (b) 3D Bi-Equivariance |

Figure 1: Illustration of bi-equivariance in 2D (left) and 3D (right). The place action, $a' = g_2 a g_1^{-1}$, is symmetric with respect to both the orientation of the object to be picked, $g_1$, and the orientation of the place target, $g_2$.

Recent works have made great progress in extending action-centric manipulation to the full SE(3) action space. Lin et al. (2023) synthesize different views and apply 2D Transporter Networks on those views to realize 3D pick-place. C2FARM (James et al., 2022) represents the workspace with multi-resolution voxel grids and infers the translation in a coarse-to-fine fashion. Transformer based methods like PerAct (Shridhar et al., 2023) first tokenize the voxel grids together with a language description of the task and learns a language-conditioned policy with Perceiver Transformer (Jaegle et al., 2021). RVT (Goyal et al., 2023) projects the 3D observation onto orthographic images and uses the generated dense feature map of each image to generate 3D actions. However, they all require a large number of expert demonstrations to learn even simple 3D pick and place skills. Compared with the previous works, our proposed architecture is an end-to-end method that demonstrates a significant increase in sample efficiency in both 2D and 3D pick-place tasks.

**Symmetries and Robot Learning.** Robot learning methods can benefit significantly by leveraging the underlying symmetry in the task. (Zeng et al., 2018; Morrison et al., 2018) show the translational equivariance of Fully Convolutional Network can improve learning efficiency for manipulation tasks. Recent works explore the use of equivariant networks (Cohen & Welling, 2016; 2017; Weiler et al., 2018; Weiler & Cesa, 2019; Cesa et al., 2021) to embed 2D rotational symmetries, resulting in a dramatic improvement in sample efficiency. Wang et al. (2021); Zhu et al. (2022) encode $SO(2)$ rotational symmetry in manipulation policies and demonstrate better on-robot grasp learning. (Wang et al., 2022b;a; Jia et al., 2023; Nguyen et al., 2023) exploit $SO(2)$ and $O(2)$ symmetries to solve multi-step manipulation tasks with a closed-loop policies. (Huang et al., 2022; 2023a) analyze the bi-equivariant symmetry of pick and place on the 2D rotation groups. However, they are limited to 2D action spaces.

Several works have attempted to utilize $SO(3)$ symmetry in robot learning. Neural Descriptor Fields (NDF) (Simeonov et al., 2022) uses Vector Neurons (Deng et al., 2021) to generate $SO(3)$-equivariant key point descriptors to define the object's pose for pick and place tasks. However, NDF and its variations (Chun et al., 2023; Huang et al., 2023b) require well-segmented point clouds and pre-trained descriptor networks. TAX-Pose (Pan et al., 2023) generates $SO(3)$-invariant dense correspondences for pick-place tasks, but is limited to reasoning over two objects. Equivariant Descriptor Field (Ryu et al., 2022) achieves bi-equivariance by encoding the $SO(3)$-equivariant point features with Tensor Field Network (Thomas et al., 2018) and SE(3) Transformer (Fuchs et al., 2020). However, it requires many samples in SE(3) to train and test the energy-based model. In comparison, our proposed method generalizes bi-equvariance to both 2D and 3D manipulation pick-and-place problems, infers the pick-and-place distribution over the entire action space with a single pass, and utilizes convolution in Fourier space to improve computation efficiency.

## 3 BACKGROUND

We provide some background on symmetry and groups, which are used in our method. Please see Appendix C for a more thorough mathematical introduction.

**Groups and Representations.** In this work, we are primarily interested in the special Euclidean group $SE(d) = SO(d) \ltimes \mathbb{R}^d$ which includes all rotations and translations of $\mathbb{R}^d$. The discrete $SO(2)$ subgroup $C_n = \{\text{Rot}_\theta : \theta = 2\pi i/n, 0 \le i < n\}$ contains rotations by angles which are multiples of $2\pi/n$. The icosahedral rotation group $I_{60}$ and octahedral rotation group $O_{24}$ are finite subgroups of the group $SO(3)$ which give the orientation preserving symmetries of the icosohedron and octahedron and contain 60 rotations and 24 rotations, respectively.

An $n$-dimensional *representation* $\rho\colon G \to \text{GL}_n$ of a group $G$ assigns to each element $g \in G$ an invertible $n \times n$-matrix $\rho(g)$ where for all $g, g' \in G$ the matrices satisfy $\rho(gg') = \rho(g)\rho(g')$. Different representations of $SO(d)$ or its subgroups describe how different signals are transformed under rotations. We consider several examples. The *trivial representation* $\rho_0\colon SO(d) \to \text{GL}_1$ assigns $\rho_0(g) = 1$ for all $g \in G$, i.e. there is no transformation under rotation. *The standard representation* $\rho_1$ of $SO(d)$ assigns each group element its standard $d \times d$ rotation matrix. For finite groups $G$, *the regular representation* $\rho_{\text{reg}}$ acts on $\mathbb{R}^{|G|}$. Label a basis of $\mathbb{R}^{|G|}$ by $\{e_g : g \in G\}$, then $\rho_{\text{reg}}(g)(e_h) = e_{gh}$. Both $SO(2)$ and $SO(3)$ have *irreducible representations*, i.e. representations with no non-trivial fixed subspaces, indexed by non-negative integers $k \in \mathbb{Z}_{\ge 0}$. The irreducible representations of $SO(3)$ are known as Wigner D-matrices, have dimension $(2k+1) \times (2k+1)$

and are denoted $D^k$. The irreducible representation $\rho_k$ of SO(2) is given by $2 \times 2$ rotation matrices $\rho_k(\theta) = \mathrm{Rot}_{k\theta}$.

**Steerable Feature Maps.** We consider all features to be steerable feature vector fields $f : \mathbb{R}^d \to \mathbb{R}^c$, which assign a feature vector $f(\mathbf{x}) \in \mathbb{R}^c$ to each position $\mathbf{x} \in \mathbb{R}^d$. An element $g \in \mathrm{SO}(d)$ acts on a steerable $\rho$-field $f$ by acting on both the base space $\mathbb{R}^d$ by rotating the pixel or voxel positions and on the fiber space $\mathbb{R}^c$ (a.k.a., channel space) by some representation $\rho$. We define the action of $g$ via $\rho$ on $f$ by $\mathrm{Ind}_\rho(g)(f)(\mathbf{x}) = \rho(g) \cdot f(\rho_1(g)^{-1}\mathbf{x})$. We denote the base space action alone by $(\beta(g)f)(\mathbf{x}) = f(\rho_1(g)^{-1}\mathbf{x})$. Note that $\beta = \mathrm{Ind}_{\rho_0}$.

***G*-Equivariant Mappings.** A mapping $F : X \to Y$ is *equivariant* to a group $G$ acting on $X$ by $\mathrm{Ind}_{\rho_X}$ and $Y$ by $\mathrm{Ind}_{\rho_Y}$ if it intertwines the two group actions $\mathrm{Ind}_{\rho_Y}(g)F[f] = F[\mathrm{Ind}_{\rho_X}(g)f]$, $\forall g \in G, f \in X$. Equivariant linear mappings, i.e. intertwiners, between spaces of steerable feature fields are given by convolution with *G-steerable kernels* (Weiler et al., 2018; Jenner & Weiler, 2021). Assume the input field type transforms as $\mathrm{Ind}_{\rho_{\mathrm{in}}}$ and the output field type as $\mathrm{Ind}_{\rho_{\mathrm{out}}}$. Then by Cohen et al. (2019), Theorem 3.3, convolution with the kernel $K : \mathbb{R}^d \to \mathbb{R}^{d_{\mathrm{out}} \times d_{\mathrm{in}}}$ is $G$-equivariant if and only if its satisfies the *steerability constraint*, $K(g \cdot x) = \rho_{\mathrm{out}}(g)K(x)\rho_{\mathrm{in}}(g)^{-1}$. A complete characterization and explicit parametrization of steerable kernels is given in Lang & Weiler (2021).

**SO(*d*) Fourier Transformation.** Signals defined over the group $\mathrm{SO}(d)$ can be decomposed as limits of linear combinations of complex exponential functions (for SO(2)) or Wigner D-matrices (for SO(3)). We refer to the Fourier transform that maps $\mathrm{SO}(d)$-signals to the coefficients of the basis functions as $\mathcal{F}^+$ and the inverse Fourier transform as $\mathcal{F}^{-1}$. For a more in depth discussion of the Fourier transform, we refer the reader to Appendix A.3.

## 4 METHOD

### 4.1 PROBLEM STATEMENT

This paper focuses on behavior cloning for robotic pick-and-place problems. Given a set of expert demonstrations that contain a sequence of one or more observation-action pairs $(o_t, a_t)$, the objective is to learn a policy $p(a_t|o_t)$ where the action $a_t = (a_{\mathrm{pick}}, a_{\mathrm{place}})$ has pick and place components and the observation $o_t$ describes the current state of the workspace. In 2D manipulation tasks, $o_t$ is in the format of a top-down image, and in 3D manipulation tasks, $o_t$ is a voxel grid. Our model factors the policy $p(a_t, o_t)$ as

$$p(a_t|o_t) = p(a_{\mathrm{place}}|o_t, a_{\mathrm{pick}})p(a_{\mathrm{pick}}|o_t)$$

where $p(a_{\mathrm{pick}}|o_t)$ and $p(a_{\mathrm{place}}|o_t, a_{\mathrm{pick}})$ are parameterized as the output of two separate neural networks. The pick action $a_{\mathrm{pick}} \in \mathrm{SE}(d)$ and place action $a_{\mathrm{place}} \in \mathrm{SE}(d)$ are decomposed in terms of translation and orientation $(T, R) \in \mathrm{SE}(d)$, where $T$ is pixel coordinates $(u, v)$ in 2D or voxel coordinates $(i, j, k)$ in 3D[1]. The rotational part of the action $R \in \mathrm{SO}(d)$ denotes the gripper orientation, which is a planar rotation $\mathrm{Rot}_\theta$ in 2D tasks and a three-dimensional rotation $R \in \mathrm{SO}(3)$ in 3D tasks. The pick rotation $R_{\mathrm{pick}} \in \mathrm{SO}(d)$ is defined with respect to the world frame and $R_{\mathrm{place}} \in \mathrm{SO}(d)$ is the relative rotation between the pick pose and place pose.

### 4.2 SE(*d*)-EQUIVARIANT PICK

We first analyze the symmetry of the robotic pick task in $\mathrm{SE}(d)$, where $d = \{2, 3\}$ indicates 2D or 3D picking. Then, we present our pick network which realizes this symmetry.

**Symmetry of the Pick Action $f_{\mathbf{pick}}$.** The pick network takes an input observation $o_t$ and outputs the pick pose probability distribution over $\mathrm{SE}(d)$, i.e., $f_{\mathrm{pick}} : o_t \mapsto p(a_{\mathrm{pick}}|o_t)$. We represent $p(a_{\mathrm{pick}}|o_t)$ as a steerable field $\mathbb{R}^d \to \{SO(3) \to \mathbb{R}\}$ in which the base space determines the pick location and the fiber space (a.k.a., channel space) encodes the distribution over pick orientations. The distribution $p : \mathrm{SO}(3) \to \mathbb{R}$ transforms by $(\rho_L(g)p)(h) = p(g^{-1}h)$ where the subscript $L$ denotes the left-action of $g^{-1}$ corresponding to post-composing with the rotation. A consistent pick network

---

[1]Each pixel or voxel corresponds to a unique $(x, y, z)$ spatial coordinate in the workspace

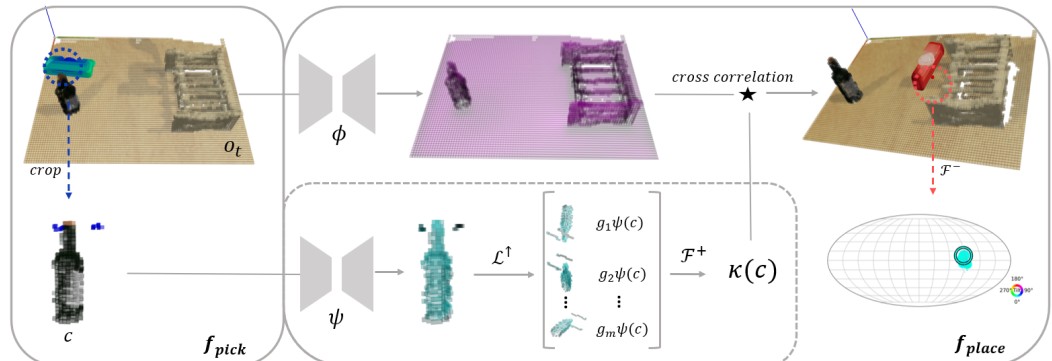

Figure 2: Architecture of FOURTRAN. $f_{\text{pick}}$ first detects a task-appropriate pick pose. The crop $c$ centered at the pick location is fed to network $\psi$. The lift operation generates a stack of rotated features and Fourier transformation $\mathcal{F}^+$ is applied to the channel space of the feature to output the dynamic kernel $\kappa(c)$. The cross correlation is conducted in Fourier space.

should satisfy the pick symmetry relation,

$$\forall g \in \text{SE}(d), \quad f_{\text{pick}}(g \cdot o_t) = \text{Ind}_{\rho_L}(g) f_{\text{pick}}(o_t) \tag{1}$$

Equation 1 illustrates the underlying symmetry of robotic picking problem, i.e., if there is a transformation $g \in \text{SE}(d)$ on $o_t$ (RHS of Equation 1), the pick pose distribution $p_{\text{pick}}$ should transform accordingly by $\text{Ind}_{\rho_L}$. Specifically, the action $\beta(g)$ on the base space rotates the pick location and the fiber action $\rho(g)$ transforms the pick orientation.

**Pick Symmetry Constraints.** To construct $f_{\text{pick}}$ satisfying Equation 1, we parameterize it with equivariant convolutional layers. The pick network $f_{\text{pick}}$ takes the input observation $o_t$ as a $\rho_0$ field and outputs a $\rho_{\text{irrep}}$ field where $\rho_{\text{irrep}}$ represents the direct sum of irreducible representations giving the truncated Fourier space representation of functions over $\text{SO}(d)$. To be specific, $f_{\text{pick}}$ can be encoded as an equivariant U-Net that takes the observation and generates the coefficients of $\text{SO}(d)$ basis functions. As a result, $f_{\text{pick}}$ generates an un-normalized distribution over $\text{SO}(3)$ above each voxel coordinate or over $\text{SO}(2)$ above each pixel coordinate. The pick pose distribution over $\text{SE}(d)$ is calculated by jointly normalizing the signal on translation and orientation. The best pick pose $\bar{a}_{\text{pick}}$ can be evaluate by $\bar{a}_{\text{pick}} = \arg\max p(a_{\text{pick}}|o_t)$.

## 4.3 $\text{SE}(d) \times \text{SE}(d)$-EQUIVARIANT PLACE

We first clarify the coupled symmetries inside place tasks. Then, we present how FOURTRAN realizes the solution.

**Symmetry of the Place Action $f_{\text{place}}$.** After the pick action $\bar{a}_{\text{pick}}$ is determined, the place network $f_{\text{place}}$ infers the place action $a_{\text{place}}$ to transport the object to be grasped to the target placement. We assume the object does not move or deform during grasping so that $a_{\text{pick}}$ may be geometrically represented by an image or voxel patch centered on the pick location[2], as shown in Figure 2. Our place network is described

$$f_{\text{place}} : (c, o_t) \mapsto p(a_{\text{place}}|o_t, a_{\text{pick}}) \tag{2}$$

where $p(a_{\text{place}}|o_t, a_{\text{pick}})$ denotes the probability that the object grasped at $a_{\text{pick}}$ in scene $o_t$ should be placed at $a_{\text{place}}$.

Since the place action is conditioned on the pick action, a consistent place network $f_{\text{place}}$ should satisfy the following bi-equivariance constraint. For $f : \text{SO}(3) \rightarrow \mathbb{R}$, denote $(\rho_R(g)f)(h) = f(hg^{-1})$ the right action on orientation distributions corresponding to pre-rotation by $g^{-1}$. The place network $f_{\text{place}}$ should be $\text{SE}(d) \times \text{SE}(d)$-equivariant:

$$\forall g_1, g_2 \in \text{SE}(d), \quad f_{\text{place}}(g_1 \cdot c, g_2 \cdot o_t) = \text{Ind}_{\rho_L}(g_2)\rho_R(g_1^{-1}) f_{\text{place}}(c, o_t) \tag{3}$$

Equation 3 states that if $g_1 \in \text{SE}(d)$ acts on the picked object and $g_2 \in \text{SE}(d)$ acts on the observation $o_t$ that contains the placement of interest[3], the desired place action will transform accordingly.

---

[2]The pick pose can also be geometrically represented by occupied voxel.

[3]Strictly, $g_2$ acts on $o_t \backslash c$, the observation excluding the patch, since the picked object is transformed by $g_1$.

Recalling $\mathrm{Ind}_{\rho_L}$ acts on the location and orientation, the place orientation $R_{\mathrm{place}}$ will be transformed to $\rho_1(g_2)R_{\mathrm{place}}\rho_1(g_1^{-1})$, and the place location $T_{\mathrm{place}}$ will be transformed to $\rho_1(g_2)T_{\mathrm{place}}$.

**Realizing place symmetry: Solution.** In order to design a network architecture $f_{\mathrm{place}}$ which satisfies Equation 2, we follow the previous works (Zeng et al., 2021; Huang et al., 2022) which treat the placing as a template matching problem and encode $f_{\mathrm{place}}$ with two separate functions $\phi$ and $\kappa$ to process $o_t$ and $c$ respectively. Then $p(a_{\mathrm{place}}|o_t, a_{\mathrm{pick}})$ is computed as the cross-correlation between $\kappa(c)$ and $\phi(o_t)$,

$$f_{\mathrm{place}}(c, o_t) = \kappa(c) \star \phi(o_t) \tag{4}$$

where $\star$ denotes the cross correlation. Specifically, $\kappa(c)\colon \mathbb{R}^d \to \mathbb{R}^{d_{\mathrm{out}} \times d_{\mathrm{in}}}$ is a dynamic kernel and $\phi(o_t)\colon \mathbb{R}^d \to \mathbb{R}^{d_{\mathrm{in}}}$ is a feature map generated from the workspace observation $o_t$.

A schematic of our proposed method FOURTRAN is shown in Figure 2. In the top branch, an encoder $\phi$ uses convolutional layers to map the input observation $o_t$ to a dense feature map. Both $o_t$ and $\phi(o_t)$ are considered $\rho_0$-fields. In the bottom branch, the crop $c$ is processed by $\psi$, which has the same architecture as $\phi$, to generate a dense feature map. Then, we lift $\psi(c)$ with a finite number of rotations $\tilde{G} = \{g_i \,|\, g_i \in \mathrm{SO}(d)\}_{i=1}^m$ to generate a stack of rotated feature maps. Specifically, we define $\mathcal{L}^{\uparrow \tilde{G}}$ as

$$\forall x \in \mathbb{R}^n, g_i \in \tilde{G} \quad \mathcal{L}^{\uparrow}[f](x) = \{f(g_1^{-1}x), f(g_2^{-1}x) \cdots, f(g_m^{-1}x)\} \tag{5}$$

a stack of fully rotated signals above each pixel or voxel. We then apply a Fourier transform to the channel-space. Our dynamic kernel generator $\kappa$ is summarized as

$$\kappa(c) = \mathcal{F}^+[\mathcal{L}^{\uparrow}(\psi(c))] \tag{6}$$

where $\mathcal{F}^+$ denotes the Fourier transform in the channel space. Finally, the cross-correlation between $\kappa(c)$ and $\phi(o_t)$ is performed in the Fourier space. Appendix.A.1 shows the pseudocode of the inference step of FOURTRAN.

Representing $\mathrm{SO}(d)$ rotations in Fourier space can achieve high angular resolution and enormously save the computation load. In contrast, directly cross-correlating between a number of rotated $\psi(c)$ and $\phi(o_t)$ independently is impossible when the action space contains a large number of rotations, especially in 3D. For example, $10^\circ$ discretization along each axis in 3D will result in $36^3$ rotations, which is 46,656 in total.

**Realizing place symmetry: Theory.** We present a general solution to achieve the bi-equivariant symmetries of Equation 3 and analyze how our porposed FOURTRAN satisfies it.

**Proposition 1** *Equation 4 satisfies the bi-equivariant symmetry stated in Equation 3 if the following constraints hold:*

1. *$\psi(c)$ satisfies the equivariant property that $\psi(g \cdot c) = \beta(g)\psi(c)$,*

2. *$\phi(o_t)$ satisfies the equivariant property that $\phi(g \cdot o_t) = \beta(g)\phi(o_t)$,*

3. *$\kappa(c)\colon \mathbb{R}^d \to \mathbb{R}^{d_{\mathrm{out}} \times d_{\mathrm{trivial}}}$ is a steerable convolutional kernel with $\rho_0$-type input.*

The full proof of Proposition 1 is in Appendix A.2. Intuitively, rotations on the crop $c$ and the placement $o_t$ result in consistent transformations of the kernel $\kappa(c)$ and feature map $\phi(o_t)$. The steerability of the kernel bridges the two transformations during the cross-correlation.

FOURTRAN satisfies the three constraints listed in Proposition 1. The first and the second constraints hold since $\psi$ and $\phi$ are implemented using equivariant convolution layers Cesa et al. (2021). The third constraint is shown in Proposition 2.

**Proposition 2** *Let $f\colon \mathbb{R}^3 \to \mathbb{R}^k$ be an $\mathrm{SO}(3)$-steerable feature field. Then, the lifting operator $\mathcal{L}^{\uparrow I_{60}}(f)$ generates a $I_{60}$-steerable kernel with regular-type output and trivial input. The fiber space Fourier transformation $\mathcal{F}^+[\mathcal{L}^{\uparrow}(f)]$ is approximately an $\mathrm{SO}(3)$-steerable kernel with trivial input and output type $\bigoplus_{\ell=0}^{\ell_{max}}(2\ell + 1)D^\ell$.*

Proposition 2 states that $\kappa(c)$ in Equation 6 is a dynamic steerable kernel generator, which satisfies the third constraint in Proposition 1. The proof of Proposition 2 is derived in Appendix B.

### 4.4 SAMPLING ROTATIONS IN A COARSE-TO-FINE FASHION

Coarse to fine sampling methods (James et al., 2022) are numerically efficient importance sampling schemes that only sample in regions of dense signal. The pick network $f_{\text{pick}}$ and place network $f_{\text{place}}$ output the $\text{SO}(d)$ distribution for each pixel or voxel in the format of coefficients of $\text{SO}(d)$ basis functions. When taking the inverse Fourier transform, we can sample a finite number of rotations for each element. However, the memory requirements for the voxel grid are cubic in the resolution of the grid and therefore limit the the number of sampled rotations for each voxel. To solve this, we first sample a small number of rotations and evaluate the best coarse pick action $(T_{\text{pick}}, R_{\text{pick}})$ and best place action $(T_{\text{place}}, R_{\text{place}})$. Then, we locate the best pick and place location $T_{\text{pick}}$ and $T_{\text{place}}$ and generate higher-order fourier signals with equivariant layers from the corresponding hidden features. Finally, we conduct a second sampling to generate a large number of fine $\text{SO}(3)$ rotations for $T_{\text{pick}}$ and $T_{\text{place}}$ and calculate the best fine orientation $R^\star_{\text{pick}}$ and $R^\star_{\text{pick}}$. The robot will receive commands of $(T_{\text{pick}}, R^\star_{\text{pick}}), (T_{\text{place}}, R^\star_{\text{place}})$ in the same time step and execute the actions.

## 5 EXPERIMENTS

### 5.1 MODEL ARCHITECTURE DETAILS

In FOURTRAN, $f_{\text{pick}}$ is a single convolutional network and $f_{\text{place}}$ is composed of two equivariant convolutional networks, $\phi$ and $\psi$. We implement them as 18-layer residual networks with a U-Net (Ronneberger et al., 2015b) as the backbone. The U-Net has 8 residual blocks and each block contains two equivariant convolution layers and one skip connection. The first layer maps the trivial-representation $o_t$ to regular representation, and the last equivariant layer transforms the $\rho_{\text{reg}}$-type feature to trivial representation for $\psi$ and $\phi$ and to a direct sum of irreducible representations for $f_{\text{pick}}$. We use pointwise ReLU activations (Nair & Hinton, 2010) inside the network.

For 3D FOURTRAN, we use $O_{24}$ regular representations in the hidden layers and lift the 3D crop features with a set of 384 approximately uniformly sampled rotations of $\text{SO}(3)$. We show that this random lifting approximately generates a steerable kernel in the Appendix B.3. To decode the $\rho_{\text{irrep}}$ feature, we first coarsely sample 384 rotations for every voxel and later conduct a finer sampling of 26244 rotations on a subregion. The maximum order $\ell_{max}$ of the Winger-D matrices in the coarse and fine sampling levels is $\ell_{max} = 2$ and $\ell_{max} = 4$, respectively. In 2D FOURTRAN, we select the $C_4$ group in the intermediate layers of the three 2D convolution networks and use the $C_{90}$ group to lift $\psi(c)$. After the Fourier transform, the frequency is truncated with the max order of 37.

### 5.2 3D PICK-PLACE

3D pick-place tasks are difficult due to the large observation and action spaces. We conduct our primary experiments on five tasks shown in Figure 3 from RLbench (James et al., 2020) and compare with three strong baselines (James et al., 2022; Shridhar et al., 2023; Goyal et al., 2023).

**3D Task Description.** We choose the five most difficult tasks from James et al. (2020) to test our proposed method. ***Stack-blocks:*** It consists of stacking two blocks of the red color on the green platform. ***Stack-cups***: In stack-cups, the agent must stack two blue cups on top of the red color cup. ***Stack-wine***: The agent must grab the wine bottle and put it on the wooden rack at one of three specified locations. ***Place-cups***: The agent must place one mug on the mug holder. This is a very high precision task where the handle of the mug has to be exactly aligned with the spoke of the holder for the placement to succeed. ***Put-plate***: The agent is asked to pick up the plate and insert it between the red spokes on the dish rack. This is also a high-precision task. The different 3D tasks are shown graphically in Figure 3. Note that the object poses are randomly sampled at the beginning of each episode and the agent needs to learn to generalize to novel object poses.

**3D Baselines.** Our method is compared against three state-of-the-art baselines: ***C2FARM-BC*** (James et al., 2022) represents the scene with multi-resolution voxels and infers the next key-frame action using a coarse-to-fine scheme. ***PerAct*** (Shridhar et al., 2023) is the state-of-the-art multi-task behavior cloning agent that tokenizes the voxel grids together with a language description of the task and learns a language-conditioned policy with Perceiver Transformer (Jaegle et al., 2021). ***RVT*** (Goyal et al., 2023) projects the 3D observation onto five orthographic images and uses the dense feature map of each image to generate 3D actions.

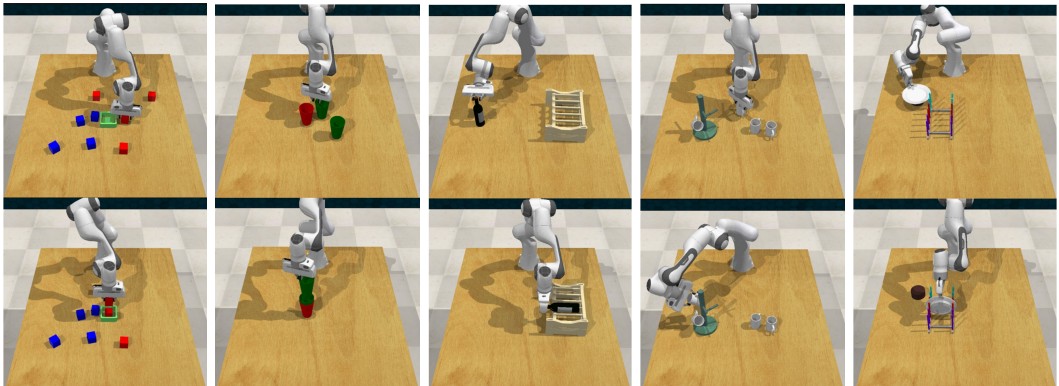

Figure 3: 3D pick and place tasks. From left to right the tasks are: Stack-blocks, Stack-Cups, Stack-Wine, Place-Cups, and Put-Plate. The top row shows the initial scene and the bottom row shows the completion state.

| Model | # demos | stack-blocks | stack-cups | stack-wine | place-cups | put-plate |
|---|---|---|---|---|---|---|
| FOURTRAN (ours) | 1 | 4 | 2.6 | 89.3 | 8 | 16 |
| FOURTRAN (ours) | 5 | 76 | 88 | 100 | 10.6 | 32 |
| FOURTRAN (ours) | 10 | **80** | **92** | **100** | **26.6** | **66.6** |
| RVT-Single | 10 | 9.3 | 13.3 | 33.3 | 1.3 | 54.6 |
| PerAct-Single | 10 | 52 | 1.3 | 12 | 1.3 | 8 |
| C2FARM-BC-Single | 10 | 36 | 0 | 1.3 | 0 | 1.3 |
| RVT-Multi | 100 | 28.8 | 26.4 | 91.0 | 4.0 | - |
| PerAct-Multi | 100 | 36 | 0 | 12 | 0 | - |
| C2FARM-BC-Multi | 100 | 0 | 0 | 8 | 0 | - |
| Discrete Expert | - | 100 | 92 | 100 | 90.6 | 65.3 |

Table 1: Performance comparisons on RL benchmark. Success rate (%) on 25 tests v.s. the number of demonstration episodes (1, 5, 10) used in training. Even with only 5 demos, our method is able to outperform existing baselines by a significant margin.

**Training and Metrics.** We train our method with $\{1, 5, 10\}$ demonstrations and train the baselines with 10 demonstrations on each task individually. The single-task versions of the baselines are denoted as '-Single'. All methods are trained for 15K SGD steps, and we evaluate them on 25 unseen configurations every 5K steps. Each evaluation is averaged over 3 evaluation seeds, and we report the best evaluation across the training process. In favor of the baselines, we also include the results of multi-task versions of the baselines trained on 16 different tasks of RLbench with 100 demonstrations per task from Goyal et al. (2023), denoted as '-Multi'. Please note put-plate task is not covered in Goyal et al. (2023). To measure the effects of discretization error and path planning, we also report the expert performance in the discrete action space used by our method.

**3D Results.** We report the results of all methods in Table 1. Several conclusions can be drawn from Table 1: 1) FOURTRAN significantly outperforms all baselines trained with 10 demos on all the tasks. 2) For tasks with a high-precision requirement, e.g., *stack-cups*, FOURTRAN keeps a high success rate while all the baselines fail to learn a good policy. 3) FOURTRAN achieves better sample efficiency, and with $\{1, 5\}$ demonstrations, it can outperform baselines trained with hundreds of demonstrations.

## 5.3 2D PICK-PLACE

We further evaluate the ability of FOURTRAN to solve precise pick-place tasks in 2D where the action space is $(u, v, \theta)$, i.e., $x, y$ translations and top-down rotation. We adopt three tasks shown in Figure 4 from the Ravens Benchmark (Zeng et al., 2021).

**2D Task Description.** *block-insertion:* The agent must pick up an L-shape block and place it into an L-shaped fixture; *assembling-kits:* The agent needs to pick 5 shaped objects (randomly sampled with replacement from a set of 20) and fit them to corresponding silhouettes on a board. *sweeping-piles:* The agent must push piles of small objects (randomly initialized) into a desired target goal

| Model | block-insertion-10 | | assembling-kits-10 | | sweeping-piles-10 |
|---|---|---|---|---|---|
| | $15°$ | $7.5°$ | $15°$ | $7.5°$ | |
| FOURTRAN (ours) | **100** | **100** | **86.2** | **78.0** | 99.8 |
| Equivariant Transporter | **100** | 98.0 | 85.0 | 76.0 | **100** |
| Transporter Net | **100** | 88.0 | 80.0 | 64.0 | 90.4 |

Table 2: Performance comparisons on 2D tasks. Success rate (%) on 100 tests. Results for both low-resolution ($15°$) reward and high-resolution reward ($7.5°$).

zone on the tabletop marked with green boundaries. Detailed task settings and descriptions can be found in Appendix A.8.

**2D Baselines.** We compare our method against two strong baselines. ***Transporter Net*** (Zeng et al., 2021) implements $\phi$ and $\psi$ with ResNet-43 without equivariant convolutional layers. It lifts the image crop $c$ to $C_n$ before feeding it to the $\psi$ network.

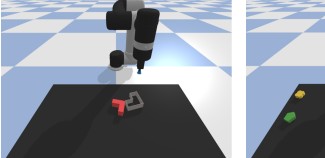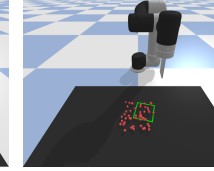

Figure 4: 2D pick and place task descriptions. Left: Block-insertion task. Center: Assembling kits task. Right: Sweeping-piles task.

***Equivariant Transporter*** (Huang et al., 2022) may be considered a variation of our proposed method with a $\rho_0$-input and $\rho_{\text{reg}}$-output steerable kernel. It is $C_n \times C_n$ equivariant. Since tasks of Ravens Benchmark use a symmetric suction gripper, all outputs of picking angle $\theta$ are equivalent. The picking angle $\theta$ is thus a nuisance variable and our pick network outputs trivial-type features (instead of regular or higher-order irreducible features).

**Training and Metrics.** We train each model with 10 expert demonstrations and measure the performance with the two reward functions. The low-resolution reward function credits the agent for translation and rotation errors relative to the target pose within $\tau = 1$cm and $\omega = 15°$ and the high-resolution reward function tightens the threshold to $\tau = 0.5$cm and $\omega = 7.5°$.

**Results.** Table 2 shows the performance of all models trained with 10 demonstrations for 10K steps. All tests are evaluated on 100 unseen configurations. First, FOURTRAN and Equivariant Transporter realize bi-equivariance and achieve a higher success rate than Transporter Net on single-step tasks and multi-step tasks. Second, as the criteria tightens from a $15°$ rotation threshold to a $7.5°$ rotation threshold, FOURTRAN maintains performance better than others. This indicates that the $\text{SO}(2) \times \text{SO}(2)$ equivariance of FOURTRAN is more precise.

## 6  CONCLUSION

In this work, we propose the FOURTRAN architecture for pick and place problems. Similar to previous pick and place methods (Ryu et al., 2022; Huang et al., 2023a), FOURTRAN leverages the coupled $\text{SE}(d) \times \text{SE}(d)$-symmetries inherent in pick-place tasks. FOURTRAN uses a novel fiber space Fourier transform method to construct a bi-equivarient architecture in a memory efficient manner. The use of Fourier space convolutions allows our architecture to process high resolution features without the need for extensive sampling. We evaluate our proposed architecture on various tasks and empirically demonstrate that our method significantly improves sample efficiency and success rate. Specifically, in the two-dimensional case, FOURTRAN achieves better performance than existing SOTA methods (Huang et al., 2023a; Zeng et al., 2021) on the Ravens benchmark. For three-dimensional pick and place, our method achieves SOTA results on a subset of tasks in the James et al. (2019) benchmark by a significant (in some cases up to two-hundred percent) margin. This empirical success establishes FOURTRAN as a powerful architecture for pick-place tasks.

One limitation of the formulation of manipulation tasks in this paper is that it relies on open-loop control and does not take path planning and collision awareness into account. Moreover, the current model is limited to a single-task setting, and extending it to a multi-task, language-conditioned equivariant agent is an important future direction. Lastly, this paper considers only robotic manipulation problems, while the bi-equivariant architecture proposed here may have uses outside of robotic manipulation. Specifically, various binding tasks in biochemistry, like rigid protein-ligand interaction (Ganea et al., 2022), and point cloud registration (Huang et al., 2021) have the same bi-equivariant symmetry as pick-place problems.

ACKNOWLEDGEMENT

Haojie Huang, Dian Wang, Xupeng Zhu, Rob Platt and Robin Walters were supported in part by NSF 1724257, NSF 1724191, NSF1763878, NSF 1750649, NSF 2107256, NSF 2134178 and NASA 80NSSC19K1474. Owen Howell was supported by the NSF-GRFP. Dian Wang was also funded by the JPMorgan Chase PhD fellowship.

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

# A    APPENDIX

## A.1    PSEUDOCODE

---

**Algorithm 1** FourTran inference

---

1: Given the observation $o_t$, pick neural network $f_{\text{pick}}$, observation and crop encoder $\phi, \psi$, coarse
   discrete SO(3) rotations $G_c$, fine discrete SO(3) rotations $G_f$
2: **if** SE($d$)-equivariant pick **then**
3:      Calculate pick logits in Fourier space: $\tilde{f}_{\text{pick}} = f_{\text{pick}}(o_t)$
4:      Calculate coarse pick distribution in spacial: $p_c(a_{\text{pick}}|o_t) = \mathcal{F}^{-}_{G_c}[\tilde{f}_{\text{pick}}]$
5:      The coarse pick pose $\bar{a}^c_{\text{pick}} = (T, R_c)$ can be evaluated by $\bar{a}^c_{\text{pick}} = \arg\max p_c(a_{\text{pick}})$
6:      The fine pick orientation $R_f$ can be evaluated at $T$ by $R_f = \arg\max \mathcal{F}^{-}_{G_f}[\tilde{f}_{\text{pick}}[T]]$
7:      $\bar{a}_{\text{pick}} = (T, R_f)$
8: **end if**
9: **if** SE($d$) $\times$ SE($d$)-equivariant place **then**
10:      Crop the observation at pick location: $c = \text{crop}(o_t, \bar{a}_{\text{pick}})$
11:      Encode observation and crop: $\phi(o_t), \psi(c)$
12:      Calculate lifted crop encoding in Fourier domain: $\kappa(c) = \mathcal{F}^{+}[\mathcal{L}^{\uparrow}_{G_c}(\psi(c))]$
13:      Calculate place logits in fourier: $\tilde{f}_{\text{place}} = \kappa(c) \star \phi(o_t)$
14:      Calculate coarse place distribution in spacial: $p_c(a_{\text{place}}|o_t) = \mathcal{F}^{-}_{G_c}[\tilde{f}_{\text{place}}]$
15:      The coarse place pose $\bar{a}^c_{\text{place}} = (T, R_c)$ can be evaluate by $\bar{a}^c_{\text{place}} = \arg\max p_c(a_{\text{place}})$
16:      The fine place orientation $R_f$ can be evaluated at $T$ by $R_f = \arg\max \mathcal{F}^{-}_{G_f}[\tilde{f}_{\text{place}}[T]]$
17:      $\bar{a}_{\text{place}} = (T, R_f)$
18: **end if**
19: return $(\bar{a}_{\text{pick}}, \bar{a}_{\text{place}})$

---

## A.2    PROOF OF PROPOSITION 1

To prove proposition 1, we begin with 2 lemmas.

**Lemma 1** *A steerable kernel* $K: \mathbb{R}^n \to \mathbb{R}^{d_{\text{out}} \times d_{\text{trivial}}}$ *satisfies*

$$\beta(g)K(x) = \rho_{\text{out}}(g^{-1})K(x) \tag{7}$$

**proof 1** *Recall that* $\rho_0(g)$ *is an identity mapping. Substituting* $\rho_{\text{in}}$ *with* $\rho_0(g)$ *and* $g^{-1}$ *with g in the steerability constraint* $K(g \cdot x) = \rho_{\text{out}}(g)K(x)\rho_{\text{in}}(g)^{-1}$ *completes the proof.*

$$\begin{aligned} \beta(g)K(x) &= K(g^{-1}x) \\ &= \rho_{\text{out}}(g^{-1})K(x)\rho_{\text{in}}(g) \\ &= \rho_{\text{out}}(g^{-1})K(x) \end{aligned}$$

**Lemma 2** *cross correlation satisfies that*

$$(\beta(g)(K \star f))(\vec{v}) = ((\beta(g)K) \star (\beta(g)f))(\vec{v}) \tag{8}$$

**proof 2** *We evaluate the left-hand side of Equation 8:*

$$\beta(g)(K \star f)(\vec{v}) = \sum_{\vec{w} \in \mathbb{Z}^2} f(g^{-1}\vec{v} + \vec{w})K(\vec{w}).$$

*Re-indexing the sum with* $\vec{y} = g\vec{w}$,

$$= \sum_{\vec{y} \in \mathbb{Z}^2} f(g^{-1}\vec{v} + g^{-1}\vec{y})K(g^{-1}\vec{y})$$

*is by definition*

$$= \sum_{\vec{y} \in \mathbb{Z}^2} (\beta(g)f)(\vec{v} + \vec{y})(\beta(g)K)(\vec{y})$$

$$= ((\beta(g)K) \star (\beta(g)f))(\vec{v})$$

*as desired.*

We first prove that if there is a rotation $u$ acting on the observation $o_t$, we have $\kappa(c) * \phi(u \cdot o_t) = \text{Ind}_{\rho_L}(u)\kappa(c) * \phi(o_t)$ and the desired place location is changed from $T$ to $\rho_1(u)T$ and the action of orientation is changed from $a$ to $ua$.

First, consider the no-rotation case without the channel-space Fourier transform.

$$\hat{\kappa}(c) * \phi(o_t) = [\beta(g_1)\psi(c), \beta(g_2)\psi(c), \cdots, \beta(g_m)\psi(c)] * \phi(o_t)$$

where $\hat{\kappa}(c) = \mathcal{L}^{\uparrow}(\psi(c))$, each rotated crop feature is cross-correlated with $\phi(o_t)$ independently and the entire output can be considered as an m-channel feature map. Assume that the best place action is found in i-th channel, i.e., $\beta(g_i)\psi(c)$ matches the dense feature map of the placement best and thus $a = g_i$.

**proof 3** *Then, consider a rotation $u$ acting on $o_t$ and assume the new best place action is found in the k-th channel*

$$\begin{aligned}
\hat{\kappa}(c) * \phi(u \cdot o_t) &= \hat{\kappa}(c) * (\beta(u)\phi(o_t)) \; \textit{Equiv. of } \phi \\
&= \beta(u)\beta(u^{-1})\hat{\kappa}(c) * \beta(u)\phi(o_t) \\
&= \beta(u)(\beta(u^{-1})\hat{\kappa}(c) * \phi(o_t)) \; \textit{lemma 2} \\
&= \beta(g)([\beta(u^{-1})\beta(g_1)\psi(c), \beta(u^{-1})\beta(g_2)\psi(c), \cdots, \beta(u^{-1})\beta(g_m)\psi(c)] * \phi(o_t)) \\
&= \beta(g)(\beta(u^{-1})\hat{\kappa}(c) * \phi(o_t)) \\
&= \beta(g)(\rho_L(u)\hat{\kappa}(c) * \phi(o_t)) \; \textit{lemma 1} \\
&= \beta(g)\rho_L(u)(\hat{\kappa}(c) * \phi(o_t)) \\
&= \text{Ind}_{\rho_L}(u)(\hat{\kappa}(c) * \phi(o_t))
\end{aligned}$$

*Note $\beta(\cdot)$ acting on the base domain while $\rho(g)$ acting on the fiber space. Assume that m is infinite and the best place action is found in the k-th channel, i.e, $\beta(u^{-1})\beta(g_k)\psi(c)$ produces the best match. We have $\beta(u^{-1})\beta(g_k)\psi(c) = \beta(g_i)\psi(c)$ and we can get:*

$$u^{-1}g_k = g_i = a \;\; \textit{since } \beta(\cdot) \textit{ is a bijective mapping}$$

*Multiplying $u$ from the left realizes that $g_k = ua$. It shows that after a rotation $u$ on the crop, the orientation component of the best place action is changed to $g_k = ua$.*

Then, we prove that if a rotation $h$ acting on the crop $c$, the desired place action to is changed from $a$ to $ah^{-1}$.

**proof 4** *Consider a rotation $h$ acting on the crop and assume the the best place action is found in the j-th channel. The place network can be evaluated as*

$$\begin{aligned}
\hat{\kappa}(\beta(h)c) * \phi(o_t) &= [\beta(g_1)\psi(\beta(h)c), \beta(g_2)\psi(\beta(h)c), \cdots, \beta(g_m)\psi(\beta(h)c)] * \phi(o_t) \\
&= [\beta(g_1)\beta(h)\psi(c), \beta(g_2)\beta(h)\psi(c), \cdots, \beta(g_m)\beta(h)\psi(c)] * \phi(o_t) \; \textit{equiv. of } \psi \\
&= \rho_R(g^{-1})\hat{\kappa}(c) * \phi(o_t) \; \textit{lemma 1 and definition of } \rho_R(g)
\end{aligned}$$

*Assume that m is infinite and the best place action is found in the j-th channel, i.e, $\beta(g_j)\beta(h)\psi(c)$ produces the best match. We have $\beta(g_j)\beta(h)\psi(c) = \beta(g_i)\psi(c)$ and since $\beta(\cdot)$ is a bijective mapping, we can get:*

$$g_j h = g_i = a$$

*Multiplying $h^{-1}$ from the right realizes that $g_j = ah^{-1}$. It shows after a rotation $h$ on the crop, the best place action is changed to $g_j = ah^{-1}$.*

Combining proof 4 and proof 3 finishes the proofs of proposition 1.

## A.3 $SO(3)$-FOURIER TRANSFORM

Let $f : SO(3) \to \mathbb{R}$ be a real valued function defined on $SO(3)$. Then, by the Peter-Weyl theorem $f$ can be decomposed as

$$\forall g \in SO(3), \quad f(g) = \sum_{\ell=0}^{\infty} \sum_{k,k'=-\ell}^{\ell} \hat{f}_{kk'}^{\ell} D_{kk'}^{\ell}(g)$$

where $D^{\ell}$ are the Wigner $D$-matrices. Wigner $D$-matrices of order $\ell$ are irreducible representations of dimension $2\ell + 1$. The Fourier transform over $SO(3)$ is defined $\mathcal{F}(f) = (\hat{f}^{\ell})_{\ell=0}^{\infty}$, where each of the $\hat{f}^{\ell}$ is a matrix of size $(2\ell + 1) \times (2\ell + 1)$. The inverse Fourier transform can be computed using the orthogonality of the Wigner $D$-matrices,

$$\hat{f}_{kk'}^{\ell} = \int_{g \in SO(3)} dg \, f(g) D_{kk'}^{\ell}(g^{-1})$$

where $dg$ denotes the $SO(3)$ Haar measure. In practice, we will truncate the $\ell$ index at some maximum value $\ell_{\max}$.

## A.4 TRAINING DETAILS

We evaluate our method on both 2D and 3D manipulation pick-place tasks. Specifically, we train a single-task policy for each task with a dataset of $n$ experiment demonstrations. Each demonstration contains one or more observation-action pairs $(o_t, \tilde{a}_{\mathrm{pick}}\tilde{a}_{\mathrm{place}})$, where $\tilde{a}_{\mathrm{pick}}$ denotes the expert pick action and $\tilde{a}_{\mathrm{place}}$ is the expert place action. We use expert actions to generate one-hot maps as the ground-truth labels for our picking model and placing model. Due to the computation load of equivariant convolutional layers on 3D voxel grids, we slightly lift the second constraint of Proposition 1 by encoding our $\phi$ with a traditional U-net (Ronneberger et al., 2015a). U-net with the long skip connection also maintains a certain amount of the $\rho_0 \mapsto \rho_0$ equivariance. Both models are trained end to end using a cross-entropy loss. The model is trained using the Adam optimizer with fixed learning rate=$1e^{-4}$. We report the training time and GPU memory requirement of 3D FOURTRAN in Table 4.

## A.5 ABLATION STUDIES

We perform two ablation studies to explore the functionality of our proposed architecture. We first replace the equivariant U-net of $\psi$ with a traditional U-net. This modification reduces the architecture to satisfy the first constraint in Proposition 1. The second ablated version of our model is that we remove the lifting and Fourier transform and directly generate the irreducible features for each element, i.e., the model is forced to learn the third constraint of Proposition 1 without prior knowledge. Table 3 shows the performances of all ablations. Comparing the first row with the second row, we find that the results are consistent. We hypothesize that the reasons are: 1) traditional U-net with the skip connections also captures the $\mathrm{trivial} \mapsto \mathrm{trival}$ equivariance and the equivariant constraints of the equivariant layers with $O_{24}$ group in FOURTRAN is not strong. 2) Data augmentation is applied to both models to learn the equivariance. Comparing the first row to the third row, the ablated model attempting to learn the coefficients of the basis function is not as well as our proposal to generate a dynamic steerable kernel. On the other hand, given the fact that lifting with a fixed number of $SO(3)$ rotations approximates the steerability of a 3D kernel, this ablated version avoids lifting to reduce the computation load. However, without representing $SO(3)$ rotations in Fourier space, it is way more expensive to evaluate a fine discrete $SO(3)$ distribution.

## A.6 MODEL COMPARISONS

We train a single-task agent for each 3D baseline on the five tasks shown in Figure 3 with the same settings presented in their paper. Note that we have {pre-pick, pick, post-pick} and {pre-place, place, post-place} where the pre-action and post-action are defined as relative to the pick and place action while the baseline predicts the action for the next keyframe without a clear line between pick and place.

| Model | stack-blocks | stack-wine | phone-on-base | put-plate |
|---|---|---|---|---|
| FOURTRAN *(ours)* | **76** | **100** | **96** | **32** |
| *Fourier Transporter w.o equiv. of $\psi$* | **76** | 96 | **96** | 24 |
| *Fourier Transporter w.o lifting* | 72 | 88 | **96** | 28 |

Table 3: Ablation Study: Performance of three variants of FOURTRAN on different RLbench tasks James et al. (2019) Each model is trained with 5 demonstrations and evaluated on 25 tests. Best performances are highlighted in bold.

| Model | Parameters (M) | Memory(Gb) | Training Time (secs/sgd step) |
|---|---|---|---|
| FOURTRAN-pick | 3.1 M | 8.5 | 1.5 |
| FOURTRAN-place | 1.6 M | 13.6 | 1.6 |
| RVT | 36 M | 12.5 | 0.46 |
| PerAct | 33 M | 12 | 1.5 |
| C2FARM | 3.6 M | 2 | 0.07 |

Table 4: Comparison of FOURTRAN (ours) architecture and existing pick-place methods. Tests were performed on NVIDIA 3090 GPU.

Table 4 compares FOURTRAN with the baseline architectures. Note that our model has about the same number of parameters as C2FARM James et al. (2020) significantly fewer parameters than the transformer-based methods PerActJaegle et al. (2021) (33 M) and RVTGoyal et al. (2023) (36 M). The equivariant 3D convolution implemented in FOURTRAN requires a relatively large GPU memory. However, FOURTRAN can produce the action distribution over the entire action space directly instead of a set of discrete rotations along each axis. As a result, FOURTRAN can easily query the backup actions if the action selected cannot be reached.

## A.7 DETAILED 3D TASK SETTINGS AND METRICS

The RLbench James et al. (2020) is implemented in CoppelaSim (Rohmer et al., 2013) and interfaced through PyRep (James et al., 2019). All experiments of RLbencg use a Franka Panda robot with a parallel gripper. Our input observations are captured from four RGB-D cameras. The captured point could is parameterized as $72 \times 96 \times 56$ voxel grid and each voxel represents a $0.94\text{cm}^3$ cube in our settings.

Figure 5 shows the expert $SO(3)$ rotation action distributions in stack-wine and put-plate tasks. The agent needs to reason about the $SO(3)$ rotation space to finish the tasks. For more information of plotting elements of $SO(3)$, please see Murphy et al. (2022).

## A.8 DETAILED 2D TASK SETTINGS AND METRICS

At the beginning of each episode, the poses of objects and placements in each task are randomly sampled in the workspace without collision. The visual observation $o_t$ is a top-down projection of the workspace with 3 simulated RGB-D cameras pointing towards the workspace. Our pixel resolution is $320 \times 160$ for the 1m $\times$ 0.5m workspace. We measure performance in the same way as it was measured in Transporter Net Zeng et al. (2021) – using a metric in the range of 0 (failure) to 100 (success). Partial scores are assigned to multiple-action tasks. For example, in the assembling kit task where the agent needs to assemble 5 objects, each successful rearrangement is credited with a score of 0.2. We report the highest validation performance during training, averaging over 100 unseen tests for each task.

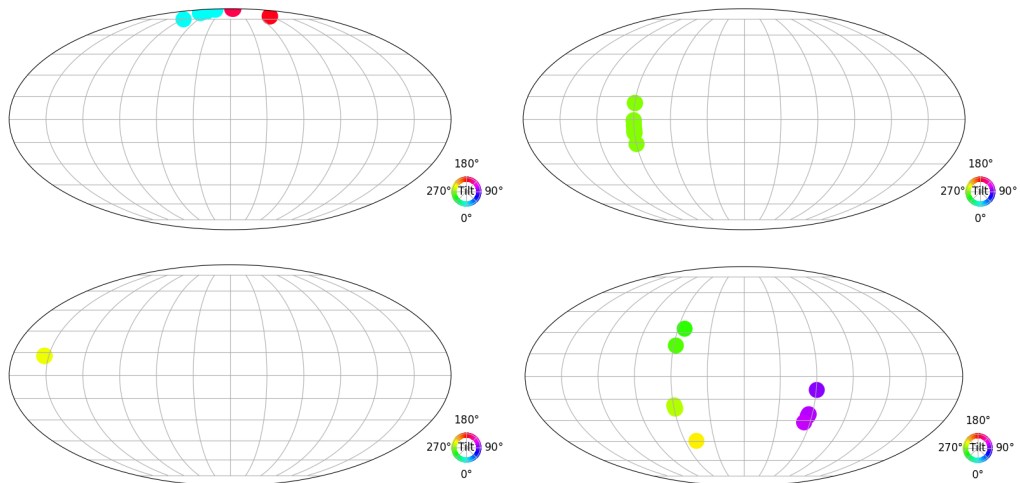

Figure 5: Visualization of expert $SO(3)$ actions from 10 demonstrations. First column: expert pick action. Second column: expert place action. First row: stack-wine. Second row: put-plate. The orientation visualization follows "YXY" convention. For more detail on plot formatting, please see Murphy et al. (2022)

## B  PROOF OF PROPOSITION 2

### B.1  THE $I_{60}$-LIFTING MAP IS A STEERABLE KERNEL WITH TRIVIAL INPUT TYPE AND REGULAR OUTPUT TYPE

The lifting map $\mathcal{L}^{\uparrow}(f)$ generates a $I_{60}$-steerable kernel with regular-type output. To see this, note that, by the definition of $\mathcal{L}^{\uparrow}$,

$$\forall x \in \mathbb{R}^d, \quad \mathcal{L}^{\uparrow}(f(x)) = \{g_1 \cdot f(x), g_2 \cdot f(x), ..., g_m \cdot f(x)\}$$

where each $g_i$ are elements of $I_{60}$. Thus, $\forall g \in I_{60}$, we have that

$$\mathcal{L}^{\uparrow}(f(g \cdot x)) = \{g_1 \cdot f(gx), g_2 \cdot f(gx), ..., g_m \cdot f(gx)\} = \{f(g_1^{-1}gx), f(g_2^{-1}gx), ..., f(g_m^{-1}gx)\}$$

Now, let $g$ be a fixed element of the icosahedral group. Left multiplication by an element of $g$ is a group homeomorphism of $I_{60}$. Let us define $g_{g(i)}$ to be the element of the sampled set $g_i$ so that

$$g_{g(i)} = g \cdot g_i$$

Let 1 be the identity element on $SO(d)$. Then, note that

$$g_{1(i)} = g_i$$

The map $g_{g(i)}$ satisfies an additional composition property. Let $g, g'$ be elements of $SO(d)$, then,

$$g_{(gg')(i)} = g_{g(g'(i))}$$

holds exactly. Thus, the map $g(\cdot) \in$ The expression for $\mathcal{L}^{\uparrow}(f)$ can be rewritten as

$$\mathcal{L}^{\uparrow}(f(g \cdot x)) = \rho(g)\{f(g_1^{-1}x), f(g_2^{-1}x)...f(g_m^{-1}x)\} = \rho(g)\mathcal{L}^{\uparrow}(f(x))$$

where the matrix $\rho$ has elements given by

$$\rho(g)_{ij} = \delta_{ig(j)}$$

Using $g_{1(i)} = g_i$, we have that the matrix $\rho$ satisfies,

$$\rho(1) = \mathbb{I}_{60}$$

where $\mathbb{I}_{60}$ is the identity matrix in 60 dimensions. Using the identity $g_{(gg')(i)} = g_{g(g'(i))}$ the matrix $\rho$ satisfies the relation

$$\rho(g)\rho(g') = \rho(gg')$$

which is the definition of a group representation. Thus, the matrix $\rho$ is exactly the permutation representation of $I_{60}$. Thus, the lifting operator $\mathcal{L}^{\uparrow}(f)$ is an $I_{60}$-steerable kernel with trivial input type and regular output type.

## B.2 FIBER SPACE FOURIER TRANSFORM

Let us suppose that the map $\mathcal{L}^{\uparrow}(f)$ is exactly a $I_{60}$-steerable kernel with trivial input type and regular output type. We can then compute the fiber space $SO(3)$-Fourier transform of $\mathcal{L}^{\uparrow}(f)$

$$\rho_{out}(g) = \bigoplus_{\ell=0}^{\ell_{\max}} m_\ell D^\ell(g)$$

where $D^\ell$ is the $\ell$-th irreducible of $SO(3)$.

Let $F^+$ be the Fourier transform in the fiber space given by

$$F^+(\mathcal{L}^{\uparrow}(f)) = \sum_{\ell=0}^{\ell_{max}} C^\ell(x) D^\ell$$

where the irreducible coefficients are given by the fiber space Fourier transform

$$C^\ell(x) = \int_{R \in SO(3)} dR \, D^\ell(R^{-1}) \rho(R) \mathcal{L}^{\uparrow}(f)(R^{-1}x)$$

## B.3 THE STOCHASTICALLY SAMPLED LIFTING MAP IS APPROXIMATELY A $SO(3)$-STEERABLE KERNEL

The stochastically sampled lifting map $\mathcal{L}^{\uparrow}(f)$ approximately generates a steerable kernel with regular-type output. To see this, note that, by the definition of $\mathcal{L}^{\uparrow}$,

$$\forall x \in \mathbb{R}^n, \quad \mathcal{L}^{\uparrow}(f(x)) = \{g_1 \cdot f(x), g_2 \cdot f(x), ..., g_m \cdot f(x)\}$$

where each $g_i$ is sampled iid from $SO(d)$. Thus, $\forall g \in SO(d)$, we have that

$$\mathcal{L}^{\uparrow}(f(g \cdot x)) = \{g_1 \cdot f(gx), g_2 \cdot f(gx), ..., g_m \cdot f(gx)\} = \{f(gg_1^{-1}x), f(g_2^{-1}gx), ..., f(g_m^{-1}gx)\}$$

Now, let $g$ be a fixed element of $SO(d)$. Left multiplication by an element of $g$ is a group homeomorphism of $SO(d)$. Let us define $g_{g(i)}$ to be the closest element of the sampled set $g_i$ so that

$$g_{g(i)} = \text{argmax}_{j=1,2,...m} ||g \cdot g_i - g_j||$$

where the norm is the geodesic distance on $SO(d)$. Let us assume that we work in the regime where the number of samples $m$ is large and

$$\max_{j=1,2,...m} ||g \cdot g_i - g_j|| \leq \frac{\epsilon}{m}$$

For iid uniform $g_i$, this property holds generically for large values of $m$. This can be proved rigorously using concentration of measure phenomena.

Let 1 be the identity element on $SO(d)$. Then, note that

$$g_{1(i)} = \text{argmax}_{j=1,2,...m} ||g_i - g_j|| = g_i$$

The map $g_{g(i)}$ satisfies an additional composition property. Let $g, g'$ be elements of $SO(d)$, then,

$$g_{(gg')(i)} = \text{argmax}_{j=1,2,...m} ||(gg') \cdot g_i - g_j|| = \text{argmax}_{j=1,2,...m} ||g(g' \cdot g_i) - g_j||$$

Thus,

$$g_{(gg')(i)} = g_{g(g'(i))}$$

holds approximately.

We may decompose the expression for $\mathcal{L}^{\uparrow}(f)(gx)$ as

$$\mathcal{L}^{\uparrow}(f(g \cdot x)) = \{f(g_{g(1)}^{-1}x), f(g_{g(2)}^{-1}x)...f(g_{g(m)}^{-1}x)\} + \textbf{error}$$

where the error term can be written as

$$\mathbf{error} = \{f(g_1^{-1}gx) - f(g_{g(1)}^{-1}x), f(g_2^{-1}gx) - f(g_{g(2)}^{-1}x), ..., f(g_m^{-1}gx) - f(g_{g(m)}^{-1}x)\}$$

if we assume that the function $f$ is $L$-Lipschitz continuous, then,

$$||f(g_k^{-1}gx) - f(g_{g(k)}^{-1}x)|| \leq L||g_k^{-1}g - g_{g(k)}^{-1}|| \leq \frac{L\epsilon}{m}$$

and so as long as $m$ is large, the error term is small (roughly $\mathcal{O}(\frac{L}{m})$). Ignoring the error term, the expression for $\mathcal{L}^\uparrow(f)$ can be rewritten as

$$\mathcal{L}^\uparrow(f(g \cdot x)) = \rho(g)\{f(g_1^{-1}x), f(g_2^{-1}x)...f(g_m^{-1}x)\} = \rho(g)\mathcal{L}^\uparrow(f(x))$$

where the matrix $\rho$ (which implicitly depends on the sampled $g_i$) has elements given by

$$\rho(g)_{ij} = \delta_{ig(j)}$$

Using $g_{1(i)} = g_i$, we have that the matrix $\rho$ satisfies,

$$\rho(1) = \mathbb{I}_m$$

where $\mathbb{I}_m$ is the $m$-dimensional identify matrix. Using the identity $g_{(gg')(i)} = g_{g(g'(i))}$ the matrix $\rho$ approximately satisfies the relation

$$\rho(g)\rho(g') = \rho(gg')$$

which is the definition of a group representation. Thus, up to approximation, the matrix $\rho$ is the permutation representation of $SO(d)$. The $g_i$ are sampled at some numerical resolution with corresponding bandwidth $\ell_{max}$. Ergo, for large $m$, the matrix $\rho$ is a good approximation to the $SO(d)$ permutation representation at bandwidth $\ell_{max}$.

## C  MATHEMATICAL BACKGROUND

We establish some notations and review some elements of group theory and representation theory. For a comprehensive review of representation theory, please see Zee (2016); Ceccherini-Silberstein et al. (2008).

### C.1  GROUP THEORY

At a high level, a group is the mathematical description of a symmetry. Formally, a group $G$ is a non-empty set combined with a associative binary operation $\cdot : G \times G \to G$ that satisfies the following properties

$$\text{existence of identity: } e \in G, \text{ s.t. } \forall g \in G, \ e \cdot g = g \cdot e = g$$
$$\text{existence of inverse: } \forall g \in G, \exists g^{-1} \in G, \ g \cdot g^{-1} = g^{-1} \cdot g = e$$

The identity element of any group $G$ will be denoted as $e$. Note that the set consisting of just the identity element $e$ is a group.

### C.2  REPRESENTATION THEORY

Let $V$ be a vector space over the field $\mathbb{C}$. A representation $(\rho, V)$ of a group $G$ consists of $V$ and a group homomorphism $\rho : G \to \text{Hom}[V, V]$. By definition, the $\rho$ map satisfies

$$\forall g, g' \in G, \ \forall v \in V, \ \rho(g)\rho(g')v = \rho(gg')v$$

Two representations $(\rho, V)$ and $(\sigma, W)$ are said to be equivalent representations if there exists an invertable matrix $U$

$$\forall g \in G, \ \ U\rho(g) = \sigma(g)U$$

The linear map $U$ is said to be a $G$-intertwiner of the $(\rho, V)$ and $(\sigma, W)$ representations. A representation is said to be reducible if it breaks into a direct sum of smaller representations. Specifically, a unitary representation $\rho$ is reducible if there exists an unitary matrix $U$ such that

$$\forall g \in G, \quad \rho(g) = U[\bigoplus_{i=1}^{k} \sigma_i(g)]U^{\dagger}$$

where $k \geq 2$ and $\sigma_i$ are smaller irreducible representations of $G$. The set of all non-equivalent representations of a group $G$ will be denoted as $\hat{G}$. All representations of compact groups $G$ can be decomposed into direct sums of irreducible representations. Specifically, if $(\sigma, V)$ is a $G$-representation,

$$(\sigma, V) = U[\bigoplus_{\rho \in \hat{G}} m_{\sigma}^{\rho}(\rho, V_{\rho})]U^{\dagger}$$

where $U$ is a unitary matrix and the integers $m_{\sigma}^{\rho}$ denote the number of copies of the irreducible $(\rho, V_{\rho})$ in the representation $(\sigma, V)$.

## C.3   WIGNER $D$-MATRICES

The fundamental representation of the rotation group in three-dimensions is given by

$$SO(3) = \{ \ R \ | \ R \in \mathbb{R}^{3 \times 3}, \ R^T R = \mathbb{I}_3, \ \det(R) = 1 \ \}.$$

It should be noted that a group is an abstract mathematical object and that the standard parameterization is a *choice*. There are multiple non-equivalent representations of the group $SO(3)$. The irreducible representations of SO(3) are known as Wigner $D$-matrices $(D^{\ell}, V^{\ell})$. The Wigner $D$-matrix of order $\ell$ is a real representation of dimension $(2\ell + 1) \times (2\ell + 1)$. Although Wigner $D$-matrices are difficult to visualize, the $\ell = 1$ representation is just the standard $3 \times 3$ matrix representation of $SO(3)$.

## C.4   PETER-WEYL THEOREM AND FIBER SPACE FOURIER TRANSFORM

The Peter-Weyl theorem (Ceccherini-Silberstein et al., 2008) states that all representations of compact groups can be decomposed into a countably infinite sets of irreducible representations. Consider the set of functions

$$\mathcal{F} = \{ \ f \ | \ f : G \to \mathbb{C} \ \}$$

of all complex valued function defined on $G$. The set $\mathcal{F}$ forms a vector space over the field $\mathbb{C}$. The group $G$ acts on vector space $\mathcal{F}$. Specifically, define the group action $\lambda : G \times \mathcal{F} \to \mathcal{F}$ as

$$\forall f \in \mathcal{F}, \ \forall g, g' \in G, \quad (\lambda_g \cdot f)(g') = f(g^{-1}g') \in \mathcal{F}$$

The action satisfies $\lambda_g \lambda_{g'} = \lambda_{gg'}$ and is a group homeomorphism. The left-regular representation of a group is defined as $(\lambda, \mathcal{F})$. The Peter-Weyl theorem Ceccherini-Silberstein et al. (2008) states that

$$(\lambda, \mathcal{F}) = U[\bigoplus_{\rho \in \hat{G}} d_{\rho}(\rho, V_{\rho})]U^{\dagger}$$

where $U$ is the unitary matrix. Thus, the left-regular representation decomposes into $d_{\rho}$ copies of each $(\rho, V_{\rho})$ irreducible. In other words, the Peter-Weyl theorem states that matrix elements of irreducible $G$-representations form an orthonormal base of the space of square-integrable functions on $G$.

## C.5   IRREDUCIBLE REPRESENTATION ORTHOGONALITY RELATIONS

Matrix elements of irreducible representations satisfy a set of orthogonality relations Zee (2016). Specifically, let $\rho$ and $\sigma$ be irreducible representations of the group $G$. Then,

$$\sum_{g \in G} \rho_{kk'}(g)\sigma(g)^{\dagger}_{nn'} = \frac{|G|}{d_{\rho}}\delta_{\rho,\sigma}\delta_{kn}\delta_{k'n'}$$

where $|G|$ is the cardinality of the group. The orthogonality relations of different irreducible representations is an analogue of harmonics of different frequencies in Fourier analysis. As an example, consider the case with $G = SO(2)$. Irreducible representations of $SO(2)$ are all one dimensional and labeled by an integer. They are all of the form

$$\rho_k(\phi) = \exp(ik\phi)$$

The orthogonality relations applied to $G = SO(2)$ then state that

$$\int_0^{2\pi} d\phi \rho_k(\phi)\rho(\phi)^\dagger_{k'} = \int_0^{2\pi} d\phi \exp(i(k-k')\phi) = 2\pi\delta_{kk'}$$

which is the standard orthogonality relation for different Fourier harmonics.

## C.6    INDUCED REPRESENTATIONS

The induced representation is a way to construct representations of a larger group $G$ out of representations of a subgroup $H \subseteq G$. Let $(\rho, V)$ be a representation of $H$. The induced representation of $(\rho, V)$ from $H$ to $G$ is denoted as $\mathrm{Ind}_H^G[(\rho, V)]$. Define the space of functions

$$\mathcal{F} = \{ \ f \ | \ f : G \to V, \ \forall h \in H, \ f(gh) = \rho(h^{-1})f(g) \ \}$$

Then the induced representation is defined as $(\pi, \mathcal{F}) = \mathrm{Ind}_H^G[(\rho, V)]$ where the induced action $\pi$ acts on the function space $\mathcal{F}$ via

$$\forall g, g' \in G, \ \forall f \in \mathcal{F}, \quad (\pi(g) \cdot f)(g') = f(g^{-1}g')$$

The induced representation was originally used in Cohen & Welling (2017) to design networks that are equivariant with respect to both rotations and translations. Induced representations can be used to change the underling group equivariance Cesa et al. (2021); Weiler & Cesa (2019); Howell et al. (2023). Specifically, induced representations can be used to design $SE(3)$-equivariant maps from $SO(3)$-equivariant maps.

