# OpenReview forum: "Fourier Transporter: Bi-Equivariant Robotic Manipulation in 3D"
_ICLR.cc/2024/Conference — ICLR 2024 poster_

### Official Review · Reviewer_vhc8 · 2023-10-31

**Soundness:** 3 good
**Presentation:** 3 good
**Contribution:** 2 fair
**Rating:** 6
**Confidence:** 4

**Summary:**

This paper introduces the fourier transporter, a neural architecture that explicitly captures the bi-equivariant relationships implicit in many pick and place tasks, that is, an object may exhibit rotational symmetry in both picking actions, and placing actions. The architecture first selects an appropriate pick pose using a network that outputs a distribution over picking actions (positions and orientations), with positions used to crop a region about the object to be placed.  This region is then lifted to form a stack of rotated features (a steerable filter) by a network to capture the rotational symmetry present in the picking action. The fourier transform of these features is applied, and used to perform cross-correlation (in the fourier domain, to allow for more efficient computation) with a feature map generated over the workspace observation to determine a distribution over placing positions.   A coarse to fine approach is used to refine the resolution of the pick and place actions, by sampling more rotations as required to refine the pick and place actions. Behaviour cloning experiments are conducted on a range of simulated 3D and 2D manipulation tasks (RLBench, Raven) and show improved success in terms of success rate as a function of training demonstrations. Ablations appear to indicate that most of the heavy work is done by the lifting operation.

**Strengths:**

The paper is well written and motivated, and does an excellent job of formalising the equivariance in robotic pick/place tasks, nicely mapping theory to practice.

The core contribution (applying the cross-correlation in the fourier domain) is a great way to reduce complexity and allow more lifting angles and finer resolution pick/place, particularly when combined with coarse to fine sampling.

Bi-equivariant networks provide a seemingly impressive boost in performance when compared to prior models that do not consider these symmetries.

**Weaknesses:**

The core weakness of this work is the strength of the contribution when compared to the equivariant transporter network proposed in Huang 2022. As far as I can tell key differences include generalising to 3D, more empirical experiments in this domain, and the implementation of the cross correlation in the fourier domain. As mentioned in this work, it is true that the Huang 2022 paper only considers SO(2), and is a subset of the general theory presented here, but more needs to be done to justify why the extension to 3D is non-trivial, particularly when it comes to the major claims of this work, greater angular resolution, computational benefits of fourier implementation, and sample efficiency.

Along these lines, I would have liked to see an explicit experiment showing clear evidence of higher angular resolution performance (beyond the 15/7.5 degree results in table 2).

No error bars are provided in experiments (Tables 1/2), so we have no indication that the results are significant. I am sure they are, but this is important for the table 2 comparison with Huang 22.

The mapping between Figure 2 and the equations in Section 4 is incomplete, and not easily followed. Not all notation is clearly defined (eg. $Ind_{\rho l}, \rho_{irrep}, h$ etc.) and equations don't immediately use the network notations ($\psi, \phi$). I recognise that much of this notation is standard in group theory, but it is not in robot learning, so there would be value in defining this. This forces the reader to make assumptions/ spend significant time interpreting the mappings between text and figures, and hurts readability. I also recognise that this is out of a desire to formalise and explain the general problem before introducing the specifics of the architecture and approach taken to address this, but the current structure of this section/ group theory jargon made this difficult to follow.

Haojie Huang, Dian Wang, Robin Walters, and Robert Platt. Equivariant Transporter Network. In Proceedings of Robotics: Science and Systems, New York City, NY, USA, June 2022.

**Questions:**

What level of rotational variation is present in the demonstrations and experiments? Is it possible to share some data on the typical distributions and tolerances in pick/place angles the tasks here require?

Could you explain table 3 in more detail - the  ablation here appears to undermine many of the choices this work makes. If a unet + data augmentation can capture many of the equivariance relations, and the lifting operation is the big contributor, then why do we need bi-equivariance? Is this simply an artifact of the test scenarios not adequately evaluating 3D equivariance symmetries?

In terms of the extension to 3D, it seems that the lifting operator introduces challenges around partial occlusions, that may be hard to learn regardless of the bi-equivariance structure. Could you comment on the general performance/ potential limitations in this regard?

---

> ### Author Response · Authors · 2023-11-21
> **Response to Reviewer vhc8**
>
> Thank you very much for the useful feedback.
>
> $\textcolor{red}{ \text{ Weaknesses: }} $
>
> **The core weakness of this work is the strength of the contribution when compared to the equivariant transporter network proposed in Huang 2022... more needs to be done to justify why the extension to 3D is non-trivial, particularly when it comes to the major claims of this work, greater angular resolution, computational benefits of Fourier implementation, and sample efficiency.**
>
> We totally disagree that this paper is not significant relative to Huang et al. 2022.
>
> 1. The generalization from the one-dimensional abelian group $SO(2)$ to the three-dimensional non-abelian group $SO(3)$ is highly non-trivial. The method proposed in (Huang 2022) cannot not be extended in three dimensions directly. (Huang 2022) requires the use of one rotation for every element of the group $G$. For subgroups of $SO(2)$, the complexity is O$(n)$, but for the case $SO(3)$, this complexity is O$(n^3)$ -- something that is totally intractable using 3D translational convolution. For example, consider that (Huang 2022) uses $C_{36}$ which has an angular resolution of $10$ degrees and requires $36$ copies of the input feature map to construct a steerable kernel. To achieve the same angular resolution in $SO(3)$, this would require copying the pick feature voxel map into $36^3/2=23228$ discrete orientations.
>
> 2. Representing the action distribution with Fourier coefficients is a significant conceptual improvement from (Huang 2022). By utilizing the Fourier Transform on groups, we can sidestep the challenges described earlier and perform memory efficient and accurate computation in three-dimensional space. It allows us to compute and represent the pick/place distribution over continuous action space. To the best of our knowledge, we are the first to represent the pick-place action distribution with Fourier coefficients.
>
> 3. The generalization into 3D is very important from a practical perspective. Many imitation learning methods reason in 3D and this paper is the first to design a Transporter-like model for this setting.
>
> **No error bars are provided in experiments (Tables 1/2), so we have no indication that the results are significant. I am sure they are, but this is important for the table 2 comparison with Huang 22**
>
> The results reported in Tables 1 and 2 are averaged over 100 trials. We have added a note to this effect in the text. We will rerun experiments and include error bars in the final version of our work.
>
> **The mapping between Figure 2 and the equations in Section 4 is incomplete, and not easily followed... the current structure of this section/group theory jargon made this difficult to follow.**
> We want our work to be readable to a wide audience, specifically researchers in robotic manipulation who are unfamiliar with group theory. For this reason, we have added additional section on mathematical background. We have added pseudocode (see new supplementary) that describes our algorithm on an intuitive level.
>
> $\textcolor{red}{ \text{ Questions: }} $
>
> **What level of rotational variation is present in the demonstrations and experiments? Is it possible to share some data on the typical distributions and tolerances in pick/place angles the tasks here require?**
>
> We have added Figure.5 in appendix to plot the $SO(3)$ distribution of expert actions on two tasks.
>
> **Could you explain table 3 in more detail - the ablation here appears to undermine many of the choices this work makes... why do we need bi-equivariance? Is this simply an artifact of the test scenarios not adequately evaluating 3D equivariance symmetries?**
>
> The ablation does not undermine our design, because both data augmentation and the lifting operation contribute to the bi-equivariance property of the network. In Proposition 1, we show three constraints that are needed for bi-equivariance; however, the second and third constraints can be implemented through different methods. Results in table 3 are not artifact of the test scenarios not adequately evaluating 3D equivariance; they evaluate different methods to achieve 3D equivariance.
>
> **In terms of the extension to 3D, it seems that the lifting operator introduces challenges around partial occlusions... Could you comment on the general performance/ potential limitations in this regard?**
>
> Occlusion in 3D is a challenge that all methods in this area must address. Our method does about as well as the baselines in this regard. Partial observation is common in 3D and affects the equivariance of our model to some extent. The crop centered at the pick location usually contains only part of the picked object (stack-wine, put-plate) and parts of neighboring objects. Our model lifts the dense feature map (dense descriptors) from the crop instead of the raw crop signal. Equivariance is built on the top of the 3D CNN and our model has the same robust reasoning and learning ability as CNN to deal with the partial observation.

---

> > ### Comment · Reviewer_vhc8 · 2023-11-21
> > **Thanks**
> >
> > Thanks for the response to my feedback. Just a note re my core weakness point around relationship to Huang 22 - I understand and agree with all the points you make here - my main comment is that this wasn't communicated strongly or explicitly enough in the original draft. This is a very important point to emphasise, since it is central to your claims. I would encourage you be as explicit as you are in your response to me in the paper, to more strongly support the value of the contribution of this work.

---

> ### Author Response · Authors · 2023-11-22
> **Response to Reviewer vhc8**
>
> Thank you for the quick response and clarification. Great point. We will definitely include the more explicit description of our contributions in the final draft!

---

### Official Review · Reviewer_sjsh · 2023-10-31

**Soundness:** 3 good
**Presentation:** 3 good
**Contribution:** 3 good
**Rating:** 6
**Confidence:** 3

**Summary:**

This paper introduces a novel approach for solving 2D and 3D pick and place tasks. The key innovation lies in leveraging Fourier transformation in fiber space to create a memory-efficient and sample-efficient bi-equivariant model. The paper provides theoretical analyses of the method and evaluates it in 2D and 3D simulation benchmarks. When compared to other methods on the RLBench (James et al. (2020)), this approach achieves a substantially higher success rate, and on the Ravens benchmark (Zeng et al., 2021), it demonstrates some improvements.

**Strengths:**

The paper shows novelty in the use of Fourier transformation in fiber space, leading to memory efficiency and enhanced sample efficiency for 3D pick and place tasks. Additionally, the proposed methods demonstrate superior performance compared to baseline approaches in select RLBench tasks.

**Weaknesses:**

While the paper demonstrates strong results on RLBench tasks, it's important to note that some tasks like "stack-blocks" and "stack-cups" primarily operate in 2D space, which may not fully reveal the strengths of the methods in 3D. It would be valuable to include additional
tasks that involve more 3D rotation angles, such as “put books on bookshelf”.

**Questions:**

In section 5.3 2D Pick-Place results, the last line: ”It indicates that the
SO(2) × SO(2) equivariance of FOURTRAN is more sensitive to rotations.
”. Does “sensitive” means more precise or prone to noise? It would be interesting to conduct separate tests with high-resolution thresholds to distinguish the impact of position error and rotation error. For example, considering parameters like τ = 1cm and ω = 7.5&deg; as well as τ = 0.5cm and ω = 15&deg;. Additionally, a box plot of the rotation error would also provide more insight into the effect of the method.


Minor issues and typos

* The last line in 3 Background: Appendix C
* The last line on page 4: “Here the action on the base space rotates the pick location and the fiber action transforms the pick orientation.” should it be: “Here the action on the base space transforms the pick location and the fiber action rotates the pick orientation”?
* Page 7:  “The different 3D tasks are shown graphically in Figure 4” should be Figure 3
* Table 3: Success rate(%) of three......
* Page 15: icosohedral -> icosahedral

---

> ### Author Response · Authors · 2023-11-21
> **Response to Reviewer sjsh**
>
> We thank the reviewer for an especially detailed and thorough review. We provide a point-by-point response to the comments and questions below:
>
> $\textcolor{red}{ \text{ Weaknesses: }} $
>
> **While the paper demonstrates strong results on RLBench tasks, it's important to note that some tasks like stack-blocks and stack-cups primarily operate in 2D space, which may not fully reveal the strengths of the methods in 3D. It would be valuable to include additional tasks that involve more 3D rotation angles, such as put books on bookshelf.**
>
> Note that three of our benchmark tasks ( 'stack-wine', 'place-cups', 'put-plate') are fully 3D and require the ability to reason about the out-of-plane actions. Specifically, in the 'stack-wine' task, the agent must rotate the wine bottle from vertical to horizontal then place it in holder. In order to complete this task, the agent must be able to understand that objects can be rotated out of the plane. Furthermore, while we agree with the reviewer that the 'stack-blocks' and 'stack-cups' tasks do not require significant out-of-plane reasoning, our action space is still defined over all poses in $SE(3)$. These experiments are important because they indicate that our general $SO(3)$ method is competitive with purely $SO(2)$ methods. We will run an additional task which is similar to the 'put books on bookshelf' task.
>
> $\textcolor{red}{ \text{ Questions: }} $
>
> **In section 5.3 2D Pick-Place results, the last line: 'It indicates that the $SO(2) \times SO(2)$ equivariance of FOURTRAN is more sensitive to rotations. ”. Does “sensitive” means more precise or prone to noise?**
>
> It means more precise.  By 'sensitive', we mean that $SO(2) \times SO(2)$-equivariant model is better able to capture high resolution rotations. We have replaced sensitive by precise in the main text. This can be seen by noting that at $\omega=7.5$ degrees our method still outperforms the 2d baselines.
>
> **It would be interesting to conduct separate tests with high-resolution thresholds to distinguish the impact of position error and rotation error. For example, considering parameters like $\tau$ = 1cm and $\omega$ = 7.5° as well as $\tau$ = 0.5cm and $\omega$ = 15°. Additionally, a box plot of the rotation error would also provide more insight into the effect of the method.**
>
> These are great suggestions that will will incorporate into the final version of the paper. We will include these results when they are finished.

---

> > ### Comment · Reviewer_sjsh · 2023-11-22
> > **Thanks**
> >
> > Thanks for the clear explanation. I look forward to the results of these additional experiments.

---

> > > ### Author Response · Authors · 2023-11-23
> > > **Response to Reviewer sjsh**
> > >
> > > We thank the reviewer for their response. We will report the results of additional tests as a table in the anonymous github (due to time constraints). Furthermore, we have added a plot of output distribution for pick and place actions (Figure 5 in appendix). Note that the outputs are not simply in-plane rotations, and the gripper can pick and place objects from a wide variety of orientations (i.e. not just grasping from the z-axis as in the stack blocks task). This shows that our model can perform manipulations over all rotations in $SO(3)$. Lastly, we wish to emphasize that $SO(2)$ equivariant methods perform poorly on the 'stack-wine', 'place-cups' and 'put-plate' tasks. Our $SO(3)$ equivariant method outperforms these methods (see table 1).
> > >
> > > We hope that this addresses the reviewers questions.

---

> > > > ### Comment · Reviewer_sjsh · 2023-12-05
> > > >
> > > > Thanks for the new rotation distribution plot. I understand the model does produce 3D rotation angles. The newly provided real robot video is an effective demonstration of the model in 3D space manipulation.

---

> ### Comment · Area_Chair_t4Me · 2023-12-04
> **[Important] Response Required to Authors' Rebuttal**
>
> Dear Reviewer sjsh,
>
> As we progress through the review process for ICLR 2024, I would like to remind you of the importance of the rebuttal phase. The authors have submitted their rebuttals, and it is now imperative for you to engage in this critical aspect of the review process.
>
> Please ensure that you read the authors' responses carefully and provide a thoughtful and constructive follow-up. Your feedback is not only essential for the decision-making process but also invaluable for the authors.
>
> Thank you,
>
> ICLR 2024 Area Chair

---

### Official Review · Reviewer_n4JL · 2023-11-01

**Soundness:** 4 excellent
**Presentation:** 2 fair
**Contribution:** 3 good
**Rating:** 8
**Confidence:** 3

**Summary:**

The paper presents a method for taking advantage of bi-equivariance found in some manipulation problems (equivariance with respect to both the pick and the place pose) for representing distributions over pick-place actions, which exist in $\textrm{SE}(3) \times \textrm{SE}(3)$ and pose sample-efficiency challenges when represented without taking advantage of symmetry. The proposed method demonstrates very strong performance on a variety of imitation learning benchmarks, particularly those requiring fine-grained control.

**Strengths:**

The argument for a bi-equivariant policy is compelling. The use of Wigner D-matrices to represent an output distribution is very clever and (to my limited knowledge of the literature) seems novel. Their use in the place network to generate fast cross-correlations for bi-equivariance is definitely novel. All theory is well presented and seems well-backed, if a little dense at times to readers less versed in differential geometry and representation theory.

Empirical results are extremely compelling. The proposed method seems to strongly outperform some relatively strong baselines on very low-data BC tasks.

**Weaknesses:**

Weaknesses mostly center around presentation: the paper contains a lot of dense jargon, which is understandable given the material but could be improved:
 - Given that the Wigner D-matrix representation and corresponding 3D Fourier transform is the key insight that allows this action representation to work, it would be worth spending some more time to describe them in more detail
 - Some pseudocode/method description would be welcome

Otherwise, further analysis of the representations introduced would be nice:
 - $\ell$
 - the number of rotations in the lifting operation

**Questions:**

Is it possible that $\textrm{SO}(2)x\mathbb{R}^3$-equivariance (2D rotational+translational) is actually more general if grasp dynamics are dependent on the object's orientation with respect to gravity?

It seems like $I_{60}$ is used in the lifting operation, but as far as I can tell there's no reason the set of rotations has to form a subgroup. Is this correct, e.g. could the granularity be increased by simply sampling more rotations (roughly evenly spaced in $\textrm{SO}(3)$) in this step?

---

> ### Author Response · Authors · 2023-11-21
> **Response to Reviewer  n4JL**
>
> We thank the reviewer for the detailed feedback.
>
> $\textcolor{red}{ \text{ Weaknesses: }} $
>
> **Given that the Wigner D-matrix representation and corresponding 3D Fourier transform is the key insight that allows this action representation to work, it would be worth spending some more time to describe them in more detail**
>
> Thanks for pointing this out. We have added an additional section on mathematical background describing the Wigner-D matrices and non-abelian Fourier transform in more detail. A full exposition of representation theory would require a textbook, but we hope that our work can be intuitively understood by reader.
>
> **Some pseudocode/method description would be welcome**
> Agreed. We have added pseudocode to the supplementary material.
>
> $\textcolor{red}{ \text{ Questions: }} $
>
> **Is it possible that equivariance (2D rotational+translational) is actually more general if grasp dynamics are dependent on the object's orientation with respect to gravity?**
>
> This is a great point. We have assumed that grasp dynamics do not play a large role in our specific settings and thus $SE(3)$ symmetry is the most relevant. In the context of a good grasp and with a sufficiently large gripper closing force, the object will not shift greatly under gravity and grasping will be $SO(3)$ invariant. In cases where gravity plays a large roll in grasp dynamics, gravity may break the full $SE(3)$ symmetry to $SO(2) \ltimes R^3$ as you suggest.  A simple way to address this issue is to simply add the gravity vector as an additional input to the $SE(3)$-equivariant model as done in related work [1].
>
> [1] Fuchs, F., Worrall, D., Fischer, V., & Welling, M. (2020). Se (3)-transformers: 3d roto-translation equivariant attention networks. Advances in neural information processing systems, 33, 1970-1981.
>
> **It seems like $I_{60}$ is used in the lifting operation, but as far as I can tell there's no reason the set of rotations has to form a subgroup. Is this correct, e.g. could the granularity be increased by simply sampling more rotations (roughly evenly spaced in ) in this step?**
>
> Correct. Randomly sampling a large number of rotations is exactly what we did in our experiments.  We consider the special case where the samples align with the subgroup $I_{60}$ in order to make a theoretical connection to previous work. When the lifting operation uses $I_{60}$, we prove that the resulting filters are an $I_{60}$--steerable kernel. When we sample random rotations uniformly, this produces an approximately $SO(3)$ steerable kernel.  We have clarified this point in the draft.

---

> ### Comment · Area_Chair_t4Me · 2023-12-04
> **[Important] Response Required to Authors' Rebuttal**
>
> Dear Reviewer n4JL,
>
> As we progress through the review process for ICLR 2024, I would like to remind you of the importance of the rebuttal phase. The authors have submitted their rebuttals, and it is now imperative for you to engage in this critical aspect of the review process.
>
> Please ensure that you read the authors' responses carefully and provide a thoughtful and constructive follow-up. Your feedback is not only essential for the decision-making process but also invaluable for the authors.
>
> Thank you,
>
> ICLR 2024 Area Chair

---

### Official Review · Reviewer_DeU9 · 2023-11-07

**Soundness:** 2 fair
**Presentation:** 2 fair
**Contribution:** 2 fair
**Rating:** 5
**Confidence:** 4

**Summary:**

The paper introduces a method called Fourier Transporter (FOURTRAN) to enhance the efficiency of training robotic agents in performing pick and place actions in 3D environments. By incorporating the bi-equivariant symmetry of the problem into a behavior cloning model, FOURTRAN utilizes a Fourier transformation to encode the symmetries of these actions independently, which enables memory-efficient construction and improves sample efficiency.

**Strengths:**

- The paper proposes FOURTRAN for leveraging bi-equivariant structure in manipulation pick-place problems in 2D and 3D.
- The paper presents a theoretical framework for exploiting bi-equivariant symmetry. It contains proofs for propositions that address the symmetry constraints and properties of the model.

**Weaknesses:**

- The current model is limited in a single-task setting, while the baseline methods are designed for multi-task purposes. I'm concerned that the comparisons may not be fair.
- It relies solely on open-loop control, disregarding path planning and collision awareness.
- The paper is not well-written and some of the terms are difficult to understand. It uses a lot of notations, but many of them are not explained.
- There are no real robot experiments.

**Questions:**

- What is **fiber space** Fourier transformation?
- In Figure 2, how do you crop the object in the scene? What if there are multiple objects?

---

> ### Author Response · Authors · 2023-11-21
> **Response to Reviewer DeU9**
>
> Please find our point-by-point response below:
>
> $\textcolor{red}{ \text{ Weaknesses: }} $
>
> **The current model is limited to a single-task setting, while the baseline methods are designed for multi-task purposes. I'm concerned that the comparisons may not be fair.**
>
> This is something we specifically addressed in Table 1 where we compared the performance of our method in the single-task setting with the baselines trained both ways, i.e. with the baselines trained in the single task setting (Baseline-Single in Table 1) and in the multi-task settings (Baseline-Multi). The single task setting gives an apples-to-apples comparison in that both methods have the same training. However, we also compare to Multi-task trained baselines since, as you point out, this was the setting they were designed for. Our method outperforms the baselines either way.
>
>
> **It relies solely on open-loop control, disregarding path planning and collision awareness.**
>
> This is not a weakness of our method. The open loop setting which we use (also known as ``keypoint'' control) has become a standard paradigm in the literature on imitation learning for robotic manipulation. All the baseline methods, Robotic View Transformer [1], PerAct [2], and Coarse to Fine [3] are open loop control methods. Additionally all baseline methods use off-the-shelf path planning algorithms.
>
>
> [1] Ankit Goyal, Jie Xu, Yijie Guo, Valts Blukis, Yu-Wei Chao, and Dieter Fox. RVT: Robotic view transformer for 3d object manipulation. In 7th Annual Conference on Robot Learning, 2023
>
> [2] Mohit Shridhar, Lucas Manuelli, and Dieter Fox. Perceiver-actor: A multi-task transformer for robotic manipulation. In Conference on Robot Learning, pp. 785–799. PMLR, 2023.
>
> [3] Stephen James, Kentaro Wada, Tristan Laidlow, and Andrew J Davison. Coarse-to-fine q-attention: Efficient learning for visual robotic manipulation via discretisation. In Proceedings of the IEEE/CVF Conference on Computer Vision and Pattern Recognition, 2022
>
>
> **The paper is not well-written and some of the terms are difficult to understand. It uses a lot of notations, but many of them are not explained.**
>
> We acknowledge that the concepts could have been presented more clearly. We have tried to address this by adding a section on mathematical background and adding pseudocode to help the reader understand our method intuitively.
>
> **There are no real robot experiments.**
>
> We agree that physical robot experiments would strengthen the paper and we are currently working on that. However, we would like to note that physical experiments are not a requirement for ICLR acceptance. Several ICLR papers [1-3] have been published without physical experiments. We are nevertheless working on physical experiments and we plan to include them in the final version of this paper. As evidence that our method works on physical hardware, we have included a video clip of our method running on a Panda robot in the real-world in the supplementary material.
>
> [1] Ryu, Hyunwoo, et al. "Equivariant Descriptor Fields: SE (3)-Equivariant Energy-Based Models for End-to-End Visual Robotic Manipulation Learning." ICLR 2023
>
> [2] Yu, Albert, and Ray Mooney. "Using Both Demonstrations and Language Instructions to Efficiently Learn Robotic Tasks." ICLR 2023
>
> [3] Li, Sizhe, et al. "Contact Points Discovery for Soft-Body Manipulations with Differentiable Physics." ICLR 2022
>
>
> $\textcolor{red}{ \text{ Questions: }} $
>
> **What is fiber space Fourier transformation?**
> The fiber space Fourier transformation computes the Fourier transformation (spectral decomposition) over the channels. This operation is performed separately at each voxel.  The use of the fiber space Fourier transform is a key component of our method and we have added an additional explanation in the mathematical background section.
>
> **In Figure 2, how do you crop the object in the scene? What if there are multiple objects?**
> We crop a $24^{3}$ voxel box centered at the pick location.  The model does not rely on any object segmentation; it simply learns to crop around the object, since this is where the pick action occurs. Figure 2 is used to explain our idea. The crop usually contains part of the picked object and parts of the neighboring objects. [2] and [3] use similar crop approaches in the 2D case.
>
> [1] Ethan Chun, Yilun Du, Anthony Simeonov, Tomas Lozano-Perez, and Leslie Kaelbling. Local neural descriptor fields: Locally conditioned object representations for manipulation. arXiv preprint
> arXiv:2302.03573, 2023
>
> [2] Haojie Huang, Dian Wang, Robin Walters, and Robert Platt. Equivariant Transporter Network.
> In Proceedings of Robotics: Science and Systems, New York City, NY, USA, June 2022. doi:
> 10.15607/RSS.2022.XVIII.007
>
> [3] Yen-Chen Lin, Pete Florence, Andy Zeng, Jonathan T Barron, Yilun Du, Wei-Chiu Ma, Anthony
> Simeonov, Alberto Rodriguez Garcia, and Phillip Isola. Mira: Mental imagery for robotic affordances. In Conference on Robot Learning, pp. 1916–1927. PMLR, 2023

---

> > ### Comment · Reviewer_DeU9 · 2023-11-22
> >
> > Thank you for your reply. Most of my concerns have been addressed and I raised my rating.

---

> > > ### Author Response · Authors · 2023-11-23
> > > **Response to Reviewer DeU9**
> > >
> > > Thank you for your kind response to our comments.  If you let us know which specific concerns still remain, we will be happy to address them.

---

### Author Response · Authors · 2023-11-21
**General Response to Reviewers**

We think this is a strong paper whose main failing has been the complexity of the notation used to describe the concepts. As a majority of the reviewers (sjsh, n4JL, vhc8) have pointed out, the empirical results presented in Table 1 are strong and significantly outperform multiple current strong baselines (e.g. PerAct and RVT) on standard benchmarks. We have revised the paper to make it more readable and have added additional ablation studies. Although this is not an an excuse, we point out that it is often the case that papers on equivariant learning are hard to penetrate due to the heavy use of group theory. That should not cloud the strong empirical and theoretical results produced by a highly novel method, i.e. one that uses the Fourier transformation in fiber space and Wigner D-matrices to represent an output distribution over $SO(3)$.

We briefly summarize the changes+additions we have made to our manuscript:

1. We have performed real-robotics experiments on a Panda manipulator with multi-view D415 depth cameras. In the supplemental we include videos that show our algorithm being used for real pick and place tasks. We hope that this illustrates the real world applicability of our method.

2. We have made significant additions to make the paper more readable. We have included in the appendix a self-contained introduction to representation theory called 'Mathematical Background' (Section A.2). We have also added a section on induced representations (section A.4) in response to the questions raised by reviewer vhc8. We hope these additional sections make the text more readable to the reader without group theory background.

3. We have added a pseudocode for our proposed Fourier Transporter method (Section A.1). We hope helps the reader understand our method at a high level.

4. We have added visualizations of the learned pick and place policies for two tasks (Figure 5 and Section A.8). We comment on the interpretation of these visualizations. Furthermore, we believe that these plots illustrate the ability of our method to generate very high angular resolution pick and place actions. For an in depth discussion of how rotation matrices are represented on mollweide plots, please see [1]

Please note that all changes to the main text and supplemental are highlighted in blue.

[1] Kieran Murphy, Carlos Esteves, Varun Jampani, Srikumar Ramalingam, and Ameesh Makadia.
Implicit-pdf: Non-parametric representation of probability distributions on the rotation manifold,
2022.

---

### Meta-Review · Area_Chair_t4Me · 2023-12-06

**Metareview:**

The paper presents a novel approach in robotic manipulation, focusing on a bi-equivariant policy using Wigner D-matrices for fast cross-correlations in 3D pick and place tasks. The average score from the reviewers is 6.25, indicating a positive reception overall. The paper's primary strengths lie in its novel theoretical underpinnings and the impressive empirical results demonstrated on low-data behavior cloning (BC) tasks. However, the paper faces challenges primarily in terms of presentation and clarity.

**Justification For Why Not Higher Score:**

The paper, while presenting novel theoretical insights and demonstrating strong empirical results, is primarily limited by several issues to be properly addressed.

**Justification For Why Not Lower Score:**

The general consensus is toward acceptance, and I agree with the reviewers.

---

### Decision · Program_Chairs · 2024-01-16

Accept (poster)